# Insulin-Like Growth Factor 2 As a Possible Neuroprotective Agent and Memory Enhancer—Its Comparative Expression, Processing and Signaling in Mammalian CNS

**DOI:** 10.3390/ijms22041849

**Published:** 2021-02-12

**Authors:** Alexander Beletskiy, Ekaterina Chesnokova, Natalia Bal

**Affiliations:** Laboratory of Cellular Neurobiology of Learning, Institute of Higher Nervous Activity and Neurophysiology of the Russian Academy of Sciences, 117485 Moscow, Russia; apbeletskiy@gmail.com (A.B.); katyachesn@gmail.com (E.C.)

**Keywords:** insulin-like growth factor 2, IGF-binding proteins, IGF1R, insulin receptor, IGF2R/CI-M6P, species-specific gene expression, CNS, learning and memory, neuroprotection

## Abstract

A number of studies performed on rodents suggest that insulin-like growth factor 2 (IGF-2) or its analogs may possibly be used for treating some conditions like Alzheimer’s disease, Huntington’s disease, autistic spectrum disorders or aging-related cognitive impairment. Still, for translational research a comparative knowledge about the function of IGF-2 and related molecules in model organisms (rats and mice) and humans is necessary. There is a number of important differences in IGF-2 signaling between species. In the present review we emphasize species-specific patterns of IGF-2 expression in rodents, humans and some other mammals, using, among other sources, publicly available transcriptomic data. We provide a detailed description of Igf2 mRNA expression regulation and pre-pro-IGF-2 protein processing in different species. We also summarize the function of IGF-binding proteins. We describe three different receptors able to bind IGF-2 and discuss the role of IGF-2 signaling in learning and memory, as well as in neuroprotection. We hope that comprehensive understanding of similarities and differences in IGF-2 signaling between model organisms and humans will be useful for development of more effective medicines targeting IGF-2 receptors.

## 1. Introduction

Insulin, insulin-like growth factors 1 and 2 (IGF-1, IGF-2), their receptors and IGF-binding proteins (IGFBPs) comprise an integrated system of signal molecules working on auto-, para- and endocrine levels and participating in multiple metabolic, trophic and protective functions, from embryonic development to homeostasis maintenance in adult organism [1,2,3,4]. Insulin is best known for its key role in glucose uptake in tissues [4]. Both insulin-like growth factors were previously considered to mediate growth hormone effects [5,6] but currently only IGF-1 is believed to perform this function [7]. The effects of IGFs are necessary for prenatal development of tissues and organs and important for their postnatal development as well. A detailed analysis shows that insulin and IGFs have complementary functions in supporting growth, differentiation, migration and metabolism of different types of cells in the developing organism [3,8].

Early studies of IGF-2 functions were mostly about its role in embryonic development, as well as its possible participation in carcinogenesis [9,10]). In the last decades, however, the role of IGF-2 in learning and memory was reported. In 2011, Chen et al. in their paper “A critical role for IGF-II in memory consolidation and enhancement” [11] showed that the hippocampal expression of Igf2 mRNA increases at 20 h after training in the passive avoidance paradigm; knockdown of Igf2 after training disrupted memory formation. Moreover, intrahippocampal injection of IGF-2 immediately after training enhanced memory retention in rats. This and some other studies provided the cause for referring to IGF-2 as “a memory enhancer” and this wording was used by various authors [11,12,13,14].

Besides its participation in the normal function of the central nervous system (CNS), IGF-2 was shown to have a role in neuropathology [15,16]. Altered IGF-2 expression was reported in various diseases. For example, it was shown that schizophrenia patients who are not taking any antipsychotic medication have a decreased serum IGF-2 concentration compared with healthy controls [17]. On the contrary, patients who had been receiving a stable dose of oral antipsychotic medications had increased serum IGF-2 level [18]. Patients with Alzheimer’s disease (AD) have decreased IGF-2 expression (compared with control subjects) in some brain regions, as was demonstrated in a number of studies using post-mortem brain samples [19,20,21]. However, in a study by Agbemenyah et al. [22] an increase of IGF2 mRNA level was observed in the prefrontal cortex of AD patients compared to the control subjects but IGF-2 protein level in these samples was not altered. Interestingly, AD patients have an increased IGF-2 concentration in the cerebrospinal fluid (CSF) [23,24] that correlates with CSF amyloid-β (Aβ_42_) levels [25]. It was shown that IGF-2 treatment restores some cognitive impairments in mouse models of schizophrenia (*Dgcr8^+/−^*) and AD (Tg2576) [20,26].

Also, increased serum IGF-2 concentration was observed in the children whose mothers drank on average at least 48 g of ethanol per day during pregnancy (the amount that may have deleterious consequences on the developing central nervous system of the fetus), compared with children from the unexposed control group at 3, 4 and 5 years of age [27].

The mechanisms regulating IGF-2 synthesis and effects are quite complicated and all these mechanisms may affect IGF-2 function in physiological and pathological processes. To begin with, Igf2 mRNA transcription depends on genomic imprinting, which is also tissue-specific. *IGF2/Igf2* gene has multiple alternative promoters that initiate synthesis of different transcripts in different tissues [28]. Igf2 transcripts can be processed by alternative splicing [29]. After the transcription, IGF2/Igf2 mRNA binds with IGF2BPs (insulin-like growth factor 2 mRNA binding proteins) that regulate its trafficking [30,31]. MicroRNAs [32,33] and long non-coding RNAs [34] can modulate IGF2/Igf2 expression as well.

Translated peptide pre-pro-IGF-2 goes through a few rounds of cleavage, generating proteoforms of different length. Another important post-translational modification is glycosylation of some IGF-2 proteoforms. Like many other growth factors, IGF-2 is processed within the Golgi apparatus and then released from the cell by exocytosis [35,36]. In the extracellular space, proteins called IGFBPs (IGF-binding proteins) bind IGF-2 with high affinity and through this IGFBPs regulate IGF-2 availability for its receptors, its life-time and other parameters [37,38,39]. 

IGF-2 performs its functions by interacting with three receptors: IGF2R (insulin-like growth factor 2 receptor), IGF1R (insulin-like growth factor 1 receptor) and IR (insulin receptor; IGF-2 interacts mostly with its IR-A isoform). In some studies, it was shown that cognitive and neuroprotective effects of IGF-2 are mediated by IGF2R [2,11,40,41,42]. This receptor was for a long time considered to function as a scavenger mediating IGF-2 degradation and its signaling activity was not recognized [6,43,44] but a growing body of evidence suggests that IGF2R is able to trigger signaling pathways [15,45,46,47]. There is also evidence of IGF-2 interaction with IGF1R during memory-related processes [48], which makes things more complicated. 

Despite the fact that a number of reviews about IGF-2 and specifically about the IGF-2 function in the CNS have been published to date, some important aspects are not thoroughly discussed in many of these reviews, leading to conflicting statements. For example, there is a discrepancy between different papers about the primary source of IGF-2 in the adult organism: while some authors claim that IGF-2 is mainly synthesized in the brain [49,50], others name the liver as the organ producing most of IGF-2 [51,52,53]. This divergency is explained by species-specific IGF-2 expression in different organs. Moreover, different species have different IGF-2 proteoform spectrum present in their serum. This difference is especially important when adult rodent IGF-2 proteoforms are compared to human ones [54] and it must be considered in translational research. Because of this, we emphasized in the present review the differences in IGF-2 transcription and processing between species.

Additionally, our review summarizes the most recent data about the participation of IGF-binding proteins in learning and memory, as well as in neuropathologies. IGFBPs present another important aspect of IGF-2-related system that should be taken into account when possible pharmacological targets for cognitive enhancement are considered. We also describe the three receptors that bind with IGF-2. We highlight IGF2R because its role in both cognitive and pathological processes is important. We review confirmed mechanisms of IGF2R signaling and provide some hypotheses about additional mechanisms in which this receptor may possibly be involved, based on literature data. 

Thus, the present work provides the information which may be important for research and development of IGF2- or IGF2R-targeting strategies for treating various neuropathologies, including aging-related dysfunctions and neurodegenerative disorders.

## 2. IGF2 mRNA Expression in Different Species

IGF-2 is called “a fetal hormone” because its expression is high in many fetal tissues, but after birth, IGF-2 expression levels in many organs are drastically decreased. Still, later in life IGF2/Igf2 mRNA expression may vary significantly in different organs depending on age. Adult expression patterns of IGF2/Igf2 are also very different between species; in particular, human IGF2 expression is very different from Igf2 expression in mouse or rat.

### 2.1. Rodents (Mouse and Rat)

Northern blotting experiments demonstrated that in adult rat, Igf2 mRNA is expressed in the CNS (brain and spinal cord) and skeletal muscles. It is absent (or present at very low levels) in liver, kidneys and spleen [55,56]. Analysis of published rat transcriptome data [57,58] shows that Igf2 expression in different organs changes greatly during postnatal development (Figure 1).

As shown on the Figure 1, during the first few weeks of rat postnatal development, Igf2 expression in the liver decreases abruptly but in the brain the level of Igf2 is more or less constant during the lifetime (but see [28] discussed below). Transcriptome analysis is partially consistent with literature data demonstrating very low expression of Igf2 in adult rat liver [55,56].

Ye et al. [28] showed that Igf2 expression in dorsal hippocampus and anterior cingulate cortex drops significantly during the first two weeks of postnatal development (1–17 days). After this, Igf2 expression continues to decrease, but more slowly.

While Igf2 expression may decrease in some parts of adult rat brain, the choroid plexus is the structure where Igf2 level is high. In the same paper [28] it was shown that Igf2 expression in the choroid plexus is roughly by 2 orders of magnitude higher compared with other examined brain regions. Apparently the choroid plexus, as well as the meninges, accounts for elevated Igf2 expression in adult rodent brain [59,60].

Choroid plexus is a thin lattice-like film enriched with blood vessels. It is located under the cortex, in the ventricles. Choroid plexus is the main source of CSF. This structure includes multiple secretory cells that produce different signal molecules, including IGF-2. During the embryonic development, choroid plexus gradually grows inside the lateral ventricles. At the same time, the level of IGF-2 in CSF increases slowly, reaching its peak at embryonic days E17–19 in rat (see Figure 3C in [61]) or E19 in mice (see Figure S3B in [61]). After this, IGF-2 concentration in CSF drops but about the time of birth it is still higher in rodent brain compared to other tissues. IGF-2 in embryonal CSF supposedly regulates corticogenesis. *Igf2* gene knockout causes changes in the brain size and the thickness of different cortical layers (see Figure 5C,G in [61]).

Interestingly, expression level of *Igf2* gene in the choroid plexus may change later with age. For example, Ziegler et al. demonstrated that relative quantity of Igf2 mRNA in the choroid plexus of adult (2 months old) mouse was significantly higher compared with that of 5 days old mouse (see Figure 3G in [62]). Analysis of data [63,64,65,66] also shows the increase of Igf2 mRNA level in the choroid plexus of adult rat compared to E15 rat embryo (Figure 2).

After IGF-2 protein is synthesized in the choroid plexus, it may not only diffuse into the CSF but also enter brain regions adjacent to lateral ventricles. In the medial region of subventricular zone (SVZ), one of the adult neurogenesis sites, high IGF-2 protein immunofluorescence was detected but Igf2 mRNA was predominantly located in the adjacent choroid plexus [62]. It was also noted that Igf2 mRNA is present in granule neurons (see Figure 3H in [62]). In addition to this, some studies demonstrate that Igf2 may be expressed by hippocampal neurons [67] and neural stem cells (NSCs) [1].

Mouse brain single-cell sequencing (Drop-seq) data shows that fibroblast-like cells, that are a component of the vasculature, have the highest Igf2 expression among all identified cell types [68,69] (see Figure 3).

### 2.2. Artiodactyla (Sheep and Cows)

In domestic cow (*Bos taurus*) calves, the highest IGF2 mRNA expression among all the studied organs is registered in liver and muscles. Expression in the brain is quite low. Compared to the fetal stage, in calves IGF2 expression decreases in all organs [70].

Using Northern blots, Delhanty and Han [71] demonstrated that in sheep (*Ovis aries*), IGF2 expression level in the choroid plexus at fetal and postnatal stages (including adulthood) is relatively stable, with no drastic changes during the lifetime. Other studied brain regions of the sheep (cortex, hypothalamus, cerebellum) do not express IGF2 mRNA. IGF2 expression in sheep kidneys, muscles and some other organs decreases with age. 

### 2.3. Humans and Other Primates

In humans, like in many other mammals, IGF2 mRNA expression decreases in many organs after birth. For example, in fetal liver IGF2 mRNA level was estimated to be 61 relative units, while in adult liver the same parameter was only 6 relative units (the values were normalized to that of adult human kidney) [72]. Liver is one of the primary sources of IGF-2 in adult human body. Still, in contrast to rodents, many other organs and tissues express IGF2 mRNA in humans [33,56,72].

Unlike in rodents, in humans the brain has one of the lowest IGF2 mRNA expression levels among different organs [33,73], although some brain regions express more IGF2 mRNA than others [74]. Importantly, small IGF2 mRNA quantity is observed in the human brain (excluding the choroid plexus and the meninges) even on the fetal stage [72,75]. 

Despite generally low IGF2 mRNA level in the human brain, high sensitivity methods allowed to register its dynamic change during the organism development. Analysis of publicly available transcriptome data from Human Brain Transcriptome [76,77] and BrainSpan Atlas of the Developing Human Brain [78,79] showed that during the prenatal development, cortical IGF2 levels vary greatly between different embryos. After birth, a gradual decrease of IGF2 expression in the cortex is observed (Figure 4).

Despite the low IGF2 mRNA level, IGF-2 protein is present in the human brain, in synapses in particular, but it is not exactly known if this protein is of a neuronal origin or, like in the case of murine SVZ mentioned above, is synthesized in some other tissue. The highest IGF-2 immunoreactivity was detected in adenohypophysis samples. It was by 1–2 orders of magnitude higher than in other brain regions [80].

Little is known about gene expression in human choroid plexus. McKelvie et al. [81], using Northern blots, demonstrated that maximal IGF2 mRNA expression in the brain takes place at the fetal stage in the choroid plexus and leptomeninges; in early postnatal development IGF2 expression level decreases, but stays high in the choroid plexus.

In fetal rhesus monkey (during the second and third trimesters) brain IGF2 mRNA expression is observed in the choroid plexus, meninges and capillaries but not in neuroglial or ependymal cells [82]. In two studies where human choroid plexus was examined using immunohistochemistry [83,84], IGF-2 protein was not detected in control choroid plexus samples or in samples from human immunodeficiency virus (HIV)-positive subjects. IGF-2 was nevertheless stained in choroid plexus papilloma [84].

More recent analysis of the human choroid plexus transcriptome using the microarray technique demonstrated relatively high expression of *IGF2* as compared to other genes [85,86] (Figure 5).

Taken together, analyses of literature and transcriptomic data show that IGF2/Igf2 mRNA expression is different between species. In adult rats, the brain [55,56] (and particularly, the choroid plexus [28]) is one of the primary sources of IGF-2, while adult rat liver has very low Igf2 expression [55,56]. In adult humans, IGF2 transcription level in the brain is quite low, while IGF2 is expressed more actively in the liver, fatty tissue and some other tissues [33]. IGF2 expression in adult human choroid plexus is relatively high, according to the transcriptomic data [85,86].

While measuring mRNA expression is usually more convenient than measuring protein expression and so in most sources only mRNA level was assessed to estimate the rate of IGF-2 synthesis in the specimens, it should be noted that transcription and translation rates of the same target are not always the same, so a high IGF2/Igf2 mRNA level does not necessarily mean that IGF-2 protein level in the same system is also high. We performed a comparison of IGF-2 transcription and translation rates where the data is present. (IGF-2 protein levels in different organs of human and rat are described in more detail in the section about IGF-2 protein processing).

In general, the dynamics of IGF-2 protein concentration in the serum of different species seems to correlate with IGF2/Igf2 mRNA expression in peripheral tissues (but not in the CNS).

In rat, IGF-2 protein concentration in blood abruptly decreases during the first few weeks after birth and this decrease is especially noticeable for the mature IGF-2 proteoform [54]. There is also a drastic decrease of Igf2 mRNA expression in many peripheral organs, including liver, muscles and lungs, during early postnatal development (Figure 1; [55,56]).

In humans, the concentration of IGF-2 protein in blood increases in the first few days after birth, and in healthy people IGF-2 blood concentration is more or less stable during the lifetime [90,91]. It was shown that the level of IGF2 mRNA in peripheral tissues is high in adult humans [33,56]. The difference in IGF2/Igf2 mRNA expression between adult rats and humans may be explained at least partially by the role of the P1 promoter of the human IGF2 gene that is absent in rodents [29,92,93]. The details about species-specific promoters of this gene are discussed in the next section.

## 3. *IGF2* Expression Regulation and Genomic Imprinting

*IGF2* gene expression in mammalian tissues is subject to complex regulation at the transcriptional level. Usually only one *IGF2* allele is expressed; the other allele is silenced by genomic imprinting [33,94]. In each parental chromosome, *IGF2* and *H19* genes have mutually exclusive activity. These two genes share a pool of distal enhancers. In paternal chromatin, DNA in the imprinting control region (located just 5′ to *H19*) is methylated and boundary transcription factor CTCF cannot bind, thus allowing the same enhancers to associate with *IGF2* [33]. Thus, in mammals, *IGF2* is usually expressed from the paternal chromosome, while *H19* is transcribed from the maternal one but this mechanism is not universal and there are some tissue-specific nuances (the details are discussed below).

In mice, four alternative promoters of *Igf2* gene are known. These promoters give rise to mRNAs with identical coding region but different non-coding 5′ leader exons. Human *IGF2* gene is even more complex: there are five promoters (P0–P4) [33]. Transcripts originating from some of these promoters are not subject to genomic imprinting in some tissues and this is supposed to be one of the mechanisms allowing to maintain high *IGF2* expression level in adult humans [92,95]. P1 is absent in rodents and remains active in adult human liver [93]. However, there are some conflicting data about the role of this promoter. The comparison of promoter-specific non-coding exons of different IGF2 mRNA isoforms demonstrated that P1 in human only drives the transcription of ~15% of IGF2 mRNA in the liver and P1-driven expression was not detected in other studied organs [33]. However, more recent study that used the same data, but in which the joints between non-coding and coding exons of various IGF2 mRNA isoforms were compared, showed that P1 is the predominant promoter of this gene in the human liver [29]. Still, we may suggest that mRNA expression driven from the human P1 promoter likely does not fully account for the observed species-specific differences in IGF-2 expression pattern and that some other mechanisms regulating IGF2/Igf2 mRNA expression and leading to different IGF-2 transcription and translation rates in adult rodents and humans must exist.

In adult rats, *Igf2* is expressed in brain regions as was discussed above. Ye et al. reported manyfold higher *Igf2* gene expression in choroid plexus, dorsal hippocampus, amygdala, cingulate and prelimbic cortex compared to this gene expression in liver, kidney, spleen. Considerable difference in *Igf2* genomic imprinting was found in these regions compared to non-neuronal tissues. In this study promoter P3 of rat *Igf2* gene was shown to be more active in examined brain regions, as compared to liver where promoter P2 prevailed. Moreover, Igf2 in these brain regions was shown to be transcribed mostly from maternal allele which is usually silenced by imprinting in mammals [28]. Similar data about the “inversion” of *IGF2* genomic imprinting in the brain were reported for humans, in a study where more than 1500 of primary tissue samples were examined [96].

*Igf2* expression in the CNS may be increased in response to learning-relating processes. For example, as a result of passive avoidance learning, inducible *Igf2* expression was detected in rat hippocampus [11,28] and prelimbic/infralimbic and anterior cingulate cortex [28]. There is some interesting new data that in male rats, stress exposure during the spermatogenesis may influence IGF-2 expression in the CNS of their offspring [97].

Different researchers discovered a number of transcription regulators that control *IGF2* transcription in the CNS. These include transcription factors CCAAT-enhancer-binding proteins (C/EBP) [11], nuclear factor kappa-light-chain-enhancer of activated B cells (NF-κB) [98], and transcriptional repressors Wilm’s Tumor 1 (WT1) [99], Sirtuin 6 (SIRT6) [100], histone deacetylase 4 (HDAC4) [101].

Other signal molecules influence both transcription and translation of IGF-2. For example, fibroblast growth factor 22 (FGF22) increases *Igf2* expression to stabilize CA3 synapses in mouse hippocampus [67]. Other secreted factors that regulate IGF-2 synthesis are brain-derived neurotropic factor (BDNF) [101] and estrogens [102]. IGF signaling-related genes (including *Igf2, Igf1r* [102], *Igfbp2*, *Igfbp6* [103]) are key intermediates of estrogenic effects on neuroplasticity in cortex and hippocampus. *Igf2* expression increase in the hippocampus that facilitates learning in rodents also depends on the photoperiod length [99]. In the cingulate cortex, inactivation of circadian modulator genes *Sharp1/2* causes *Igf2* expression increase and enhances cortex-dependent remote fear memory formation. However, these genes do not have similar effects in hippocampus-dependent recent fear memory formation [104]. *Igf2* expression may also be stimulated artificially by selective serotonin reuptake inhibitors administration, as was shown in experiments with murine model of depression [105].

## 4. IGF Family Protein Processing

### 4.1. Cleavage

IGF-2 is originally synthesized as a precursor protein. As many other secreted proteins, mature IGF-2 is generated by site-specific cleavage of a longer precursor molecule. Pre-pro-IGF-2 has a molecular weight about 22 kDa [106,107]. Its *N*-terminal bears a signal peptide that is removed at an early stage of processing, leading to the formation of pro-IGF-2 (about 20 kDa) [107,108]. In some sources pro-IGF-2 forms with higher molecular mass (24–26 kDa) are mentioned [36] but this may be explained by glycosylation affecting protein migration in the gel. If glycosyl residues were not removed during the sample preparation, glycosylation makes proteins to migrate in the gel during electrophoresis slower than expected based on their predicted molecular weight. Glycosylation of various IGF-2 proteoforms is discussed in detail below. 

After the removal of the signal sequence, there are a few options for targeted proteolysis of pro-IGF-2. Cleavage at different sites generates proteoforms of different molecular weights. Usually it is the 7.5 kDa protein that is considered to be the “mature” form of IGF-2 [7,54,106,109,110]. But in some studies [108,111], 14–15 kDa proteoforms are called “mature” IGF-2. Proteoforms of different weight form as a result of incomplete cleavage [106], as well as post-translational modifications like *O*-glycosylation [35,112].

Since insulin is one of the most studied members of the insulin/IGF/relaxin superfamily, the common classification of domains comprising IGF proteins is based on insulin molecular domains. To begin with, the precursors of insulin and IGF molecules have a 24–25 aa signal sequence on their *N*-terminals. It was shown that in pre-pro-insulin molecule this sequence is necessary for directing the translated polypeptide chain inside the endoplasmic reticulum (ER). Upon delivery to the ER, the signal peptide needs to be properly oriented, with its *N*-terminus facing the cytosolic side of the ER membrane. This allows the cleavage site of the signal peptide to be exposed on the luminal side of the ER membrane. After the signal peptide is removed, pre-pro-insulin becomes pro-insulin [113]. In the pro-insulin molecule, A, B and C domains are recognized. During the processing of insulin, C-peptide between A and B domains is excised by site-specific proteolysis and A and B domains are then cross-linked by two disulfide bridges, forming the mature insulin molecule [114].

Based on homology, pro-IGF-1 and pro-IGF-2 molecules are also considered to have A, B and C domains. But unlike insulin processing, these domains are not separated during IGF proteins processing. Another difference is additional domains D and E that are present in pro-IGF peptides. Domain D, together with domains A, B and C, is retained within the mature IGF molecule [7]. In the case of human IGF-2, the mature protein weighs 7.5 kDa and is found in different tissues, including blood [54] and CSF [115].

It should be noted that E domain in pro-IGF-2 is considerably longer compared to the same domain in pro-IGF-1. This domain is where a few specific proteolysis sites (that are cleaved during the mature IGF-2 formation) are located [7].

Four main secreted forms of IGF-2 are currently recognized in the serum of human and rat: pro-IGF-2 (amino acids 1-156), two forms of “big IGF-2” (1–104 and 1–87) and mature IGF-2 (1–67) (Figure 6). Pro-IGF-2 and “big IGF-2” forms, unlike the mature IGF-2, are subject to glycosylation. Because of this, on Western blots each of these proteins may look like a few distinct bands located closely to each other [54,112].

Interestingly, in the circulating blood of adult rats, unlike humans and rhesus monkeys, pro-IGF-2 is the prevailing form, while shorter forms are either absent altogether [112] or found in much smaller concentrations [54]. Still, in rat embryos and newborn pups different IGF-2 forms are found, including the mature protein [54,112]. Analysis of IGF-2 proteoform dynamics in rat postnatal development shows the peak of mature IGF-2 level at 1–5 days after birth. After this mature IGF-2 concentration gradually decreases, and this form virtually disappears by the 25th day. The absence of mature IGF-2 persists at least until the 50th day. For pro-IGF-2, the dynamic is different: there is a peak on the 15th day after birth, then the absolute pro-IGF-2 level somewhat decreases but its fraction among all IGF-2 proteoforms steadily increases, reaching nearly 100% by the 25th day; the same ratio is maintained by the 50th day. It should be noted that in pregnant rats, blood concentration of mature IGF-2 increases [54]. Analysis of published data shows that there are two main IGF-2 proteoforms found in rodent brain, 14–17 and 20 kDa [20,108,111]. The heaviest one (20 kDa) corresponds to pro-IGF-2. The other one must be one if the “big IGF-2” forms (probably glycosylated) but additional experiments are necessary to identify it.

### 4.2. Proteases Participating in the Processing of IGF-2

Which proteases are responsible for cleaving pro-IGF-2 to produce “big IGF-2” forms and mature IGF-2? Supposedly, these proteases belong to the proprotein convertase subtilisin/kexin type (PCSK) family.

Inhibition of furin (PCSK3) proteolytic activity disrupted pro-IGF-2 cleavage in cultured human embryonic kidney (HEK) cells. But the possibility of some other PCSK convertases also participating in pro-IGF-2 proteolysis cannot be ruled out, because the furin inhibitor used in this study was likely non-specific. In additional experiments, transient cotransfection with constructs encoding pro-IGF-2 and various PCSK convertases (furin, PACE4, PC6A, PC6B or LPC) in HEK cells resulted in enhanced or complete processing of the precursor for all convertases [35]. Furin treatment of recombinant pro-IGF-2 in vitro generated the “big IGF-2” form (1–104) [54] but in the experiment described above (with pro-IGF-2 and furin coexpression in cell culture) smaller proteolysis products were also detected, which may be explained by additional non-specific proteolysis. Experiments with artificial mutagenesis of *IGF2* gene demonstrated that two pro-IGF-2 cleavage sites located at positions 67 and 104 have qualitative differences caused by different amino acids within the consensus R/K-X-X-R/K sequence motif immediately preceding the cleavage site [35]. Apparently, specificity of convertases cleaving these sites depends on these amino acids.

PC4 (PCSK4) was shown to be able to perform pro-IGF-2 cleavage at Arg^68^ and Arg^104^ sites in cultured HTR8-SVneo cells (derived from human trophoblasts) [116].

Steinmetz et al. [108] suppose that in the brain, pro-IGF-2 cleavage may be realized by proprotein convertases PC2 and/or PC1. This conclusion is based on the data about PC2 enzymatic activity in the hippocampus of rats of different ages and reverse correlation observed between PC2 activity and 20 kDa/15 kDa IGF-2 proteoform ratio.

It should also be noted that after the proprotein convertase-mediated cleavage, *C*-terminal arginine residues of a cleaved fragment may be removed by carboxypeptidases E and D [116].

### 4.3. Glycosylation

Mature IGF-2 does not have glycosylation sites but E-domain of pro-IGF-2 may be glycosylated by up to four *O*-linked sugars. Amino acid residues that serve as glycosylation sites are Ser^71^, Ser^72^, Ser^75^ and Thr^139^. Various sugars used for glycosylation have *O*-glycan backbone decorated with different number of sialic acid residues [35,117].

Do different IGF-2 proteoforms have different functions and what difference does glycosylation make? Apparently, glycosylated forms are more biologically active because of their increased availability to IR and IGF1R receptors. There are some data confirming that glycosylation of pro-IGF-2 and “big IGF-2” forms decreases their affinity to the scavenging IGF-2 receptor IGF2R [117]. Glycosylation also inhibits another mechanism of IGF-2 sequestration. IGFs in blood are mostly confined to ternary protein complexes consisting of an IGF-binding protein (IGFBP3 or IGFBP5), a molecule of IGF-1 or IGF-2 and the acid-labile subunit (ALS) [37,118] (this is discussed in more detail in the next section). Glycosylated “big IGF-2” forms have decreased affinity to ALS, hindering the ternary complex formation. “Big IGF-2” forms in concentrations similar to those of the mature form may stimulate cell proliferation through IR-A, IR-B and IGF1R receptors. Supposedly the main functional difference between IGF-2 proteoforms is not the mechanisms of IR or IGF1R receptor activation, but rather the ligand availability for these receptor tyrosine kinases. The availability in this case is modified by ligand binding with proteins like ALS and IGF2R and glycosylated “big IGF-2” proteoforms were shown to be able to escape sequestration by these proteins [117]. Still, it is too early to state that these forms have exactly the same receptor activation mechanism, because there are currently very few studies of this subject. Moreover, IGF2R function is not very clear yet which makes the situation even more complex.

Thus, post-translational modifications of pre-pro-IGF-2 lead to formation of different IGF-2 proteoforms that were shown to have different affinity to the same targets. This has to be considered when studying the role of IGF-2 in normal and pathological processes in the brain and other organs [7].

Analysis of literature data in the present review shows a significant difference between human and rat adult proteoforms of IGF-2. While in humans, the main form of IGF-2 in the serum is mature IGF-2 (7.5 kDa), in rat serum pro-IGF-2 is prevailing. Also, blood IGF-2 level in human is even higher than insulin level [7] and this should be noted while considering possible clinical applications of IGF-2. Another thing that should be considered is that most IGF-2 molecules in the bloodstream and in local tissues are bound by IGF-binding proteins (IGFBPs) that significantly alter IGF-2 activity [37].

## 5. IGF-Binding Proteins

One of the mechanisms that regulate IGFs function involves changing the availability of these peptides for their receptors. While human serum has relatively high concentration of IGF-2, most of it is present in inactive form, bound to specific proteins [119,120]. These proteins (IGFBP1-6) belong to the IGF-binding protein family and have very high affinity for IGFs (K_d_ ~10^−10^ M). In addition, the study of homology between amino acid sequences of multiple proteins revealed some proteins that are classified as members of “IGFBP superfamily.” One of these proteins is IGFBP7, which has an affinity to IGFs about 2 orders of magnitude lower compared with IGFBP family proteins [121]. Despite the homology between different IGF-binding proteins and their ability to substitute for each other in knockout animals, each of these proteins has individual characteristics. For review, see [3,37,38,44,121,122,123,124]. In general, the functions of IGF-binding proteins on molecular level are as follows:Inhibition of IGF activity by binding IGFs and reducing their access to receptors. For example, in human serum the concentration of IGF-2 is higher than that of insulin and, considering the fact that IGF-2 is able to activate insulin receptors, this would lead to hypoglycemia if IGF-2 was present in its unbound form [7]. But IGFs in blood are mostly confined to ternary protein complexes formed with IGFBP3/5 and ALS [37,118] and thus are unable to activate their receptors.IGF protection from degradation (increase of their half-life) [39,125,126].Regulation of local IGF concentration: when proteases degrade IGFBPs, their affinity to IGFs decreases, leading to the release of IGFs and increase in their local concentration [37].Interaction with cell surface molecules and extracellular matrix [37].Interaction with other growth factors [127].IGF-independent functions [37,128,129,130].

Some recently published data indicate that IGFBPs may participate in both normal and pathological function of the CNS. For example, IGFBP2 is associated with biomarkers of AD [131], and elevated concentration of circulating IGFBP2 was linked with an increased risk of both all-cause dementia and AD [132].

Among all IGFBPs, IGFBP2 is the most abundant in the CSF. This protein is also highly expressed in the developing brain [133,134]. IGFBP2 is involved in the CNS development, neuronal plasticity and higher order brain functions. For more information about IGFBP2 structure and functions in the CNS, we recommend a recent detailed review by S. Khan [133].

IGFBP3 is the primary IGFBP in the circulating blood. IGFBP3 in association with ALS forms ternary complexes with IGFs. IGFBP3/ALS complex not just inhibits the binding of IGFs with their receptors but also increases the half-life of these factors [135]. IGFBP3 alone (without ALS) may form a binary complex with either IGF-2 or pro-IGF-2 [117,136]. Smaller binary complexes permeate the capillaries easily, leading to increased bioavailability of IGF in the tissues [137]. ALS binding with IGFBP-3/pro-IGF-2 binary complexes is greatly decreased (compared with IGFBP-3/mature IGF-2 complexes) [136]. Also, binary complex formation is not impeded by glycosylation of “big IGF-2” proteoforms but recruitment of ALS to ternary complex is severely hindered by this modification [117]. Dysregulation of IGF-2 proteoform composition with increased concentration of long proteoforms is typical in non-islet cell tumor induced hypoglycemia. Disturbance of the ternary complex formation by “big IGF-2” forms, leading to release and activation of IGF-2, is assumed to be important in the pathogenesis of this condition. However, “big IGF-2” forms did not disrupt ternary complexes in the serum of a healthy subject in vitro, suggesting that IGF-2 release from the ternary complex in this pathology must have other mechanisms [137].

One of the known neuropathologies that was linked to IGFBP3 malfunction is Rett syndrome. This is a major neurodevelopmental disorder, caused by mutations in the *MeCP2* gene and characterized by mental retardation and autistic behavior. MECP2 protein suppresses *IGFBP3* gene expression, so MECP2 dysfunction leads to increased expression of *IGFBP3*. Mice with artificial *Igfbp3* overexpression have phenotype somewhat similar to *Mecp2*-null mice and some features in their CNS development resemble Rett syndrome pathologies. This may confirm the role of *IGFBP3* overexpression in Rett syndrome pathogenesis [138].

On the other hand, decreased *IGFBP3* expression level may also be associated with intellectual disability. RSRC1 (Arginine And Serine Rich Coiled-Coil 1) is a protein involved in splicing regulation [139]; RSRC1 loss-of-function mutation in humans leads to drastically decreased *IGFBP3* expression in neural progenitor cells [140]. It was later confirmed that RSRC1 loss-of-function causes mild to moderate autosomal recessive intellectual disability [139]. In mice, *Ifgbp3* knock-out leads to reduction of IGF-1 level in the brain. This suggests decreased half-life of IGF-1 in the blood serum. Moreover, *Igfbp3*-null mice had multiple alterations of synaptic function: reduced number of dendritic spines, impaired function of neuronal signaling pathways, abnormal monoaminergic neurotransmission and some behavioral deviations [126].

Interestingly, decreased IGFBP3 level was also reported in the posterior hypothalamus of both narcoleptic versus control postmortem human brains and transgenic mice lacking hypocretin neurons versus wild-type mice. Moreover, IGFBP3 was found to be colocalized with hypocretin in hypothalamic neurons. Unexpectedly, artificial overexpression of human *IGFBP3* in mice decreased both pre-pro-hypocretin mRNA and hypocretin peptide content in hypothalamus and target areas. It was found that in vitro, promoter of the corresponding gene *HCRT* is suppressed by IGFBP3. In vivo, overexpression of a mutant (unable to bind IGFs) IGFBP3 form reduced hypocretin peptide content in some brain regions, suggesting both IGF-dependent and IGF-independent effects of IGFBP3 on hypocretin transmission [141].

IGF-independent functions of IGFBP3 are further confirmed by some data about IGFBP3 being able to translocate to the nucleus, where it interacts with some nuclear receptors, participates in transcription regulation and DNA damage response. Many details of IGFBP3 nuclear function are not fully understood yet. IGFBP3-mediated nuclear import of IGF-1 was also shown [128,129]. In addition, IGFBP3 is involved in the regulation of apoptosis and survival of neurons in an IGF-independent manner [130] and inhibition of IGF-1-mediated proliferation of neural progenitor cells [142].

Thus, IGFBP3 is not only a key regulator of IGFs bioavailability but is also involved in the CNS function via both IGF-dependent and IGF-independent mechanisms.

IGFBP4 is involved in nervous system development [122,143]. Analysis of published datasets shows that Igfbp4 and Igfbp5 mRNAs have distinct expression patterns in mouse hippocampus [144,145,146,147,148] (Figure 7). In situ hybridization and transcriptome sequencing revealed that in mouse brain, Igfbp4 is primarily expressed in CA1 region, while Igfbp5 mRNA is located in dentate gyrus. In rat brain, in situ hybridization also demonstrated different localization of IGFBP mRNAs [149].

Post-mortem transcriptome analysis of elderly patients demonstrated a correlation between *IGFBP5* expression in the frontal cortex and AD progression. *IGFBP5* in this research was identified within a module (a group of co-expressed genes) (see Table S8 in [150]); [151]. Further analysis of brain samples proteomes demonstrated that higher IGFBP5 protein level correlates with more rapid age-associated cognitive decline. Notably, the link between IGFBP5 expression and cognitive decline was not fully explained by AD, Lewy body and TDP-43 pathologies. This suggests that IGFBP5 may also be mediating the effects of some other neuropathologies [151]. In addition, a correlation between high IGFBP5 expression level in the cortex and declined motor function was found [152].

IGFBP7 should be discussed separately. This protein has much less affinity to IGFs compared with other IGFBPs but on the other hand, there is some data about IGFBP7 playing an important role in both normal and pathological function of the CNS.

In the study by Agis-Balboa et al. [48], the level of IGF-2 in the murine hippocampus was shown to increase an hour after the first extinction trial (that may also be considered a reminder). At the same time, IGFBP7 level did not change after the reminder but was strongly downregulated after the last extinction trial in a row (on which the freezing behavior was significantly reduced compared to the reminder trial). Intrahippocampal injection of IGFBP7 after each extinction trial disrupted fear extinction, while co-injection of IGF-2 rescued this effect. This may be considered an indirect proof that IGFBP7 interacts with IGF-2 in vivo. In the same study, the role of both IGF-2 and IGFBP7 in the regulation of adult neurogenesis and survival of immature neurons during fear extinction was confirmed.

IGFBP7 expression level was shown to be changed in AD. The study by Agbemenyah et al. [22] demonstrated that both mRNA and protein expression of IGFBP7 are increased in the prefrontal cortex of AD patients and in the hippocampus of APPPS1-21 mice (a genetic model of AD). Notably, the levels of IGFBP1-6 were not increased in these samples. Moreover, intrahippocampal IGFBP7 injection 1 h before training inhibited learning in mice in both fear conditioning and water maze paradigms.

Taken together, these data demonstrate that regulation of IGFBP7 expression is important for both memory formation and extinction and that the change in IGFBP7 level is associated with AD.

## 6. IGF-2 Receptors (IGF1R, IR, IGF2R)

IGF-2 is capable to bind with IR (insulin receptor), IGF1R (insulin-like growth factor 1 receptor) and IGF2R (insulin-like growth factor 2 receptor). IGF1R and IR belong to the receptor tyrosine kinase (RTK) superfamily [8,153,154]. The third IGF-2 target, IGF2R, is structurally different from IR and IGF1R [15,155,156]. All three receptors do not exclusively bind IGF-2 and have other ligands.

IR has two isoforms, IR-A and IR-B. IR-A (but not IR-B) binds IGF-2 with an affinity close to that of insulin [157,158]. IGF-2 binds IR-B and IGF1R with lower affinity than their primary ligands, insulin and IGF-1 respectively [157,159]. IGF2R is able to bind IGF-2 with much higher affinity than IGF-1 and has no affinity to insulin [156,160,161]; this receptor also has multiple binding sites for various ligands that are not members of insulin/IGF/relaxin superfamily [15,156,162].

IGF1R and both IR isoforms may also form dimers with each other, which makes things even more complicated. If we arranged all possible receptors by IGF-2 affinity to them, the list would most likely look like this: IGF1R > IGF2R = IR-A = IGF1R / IR-A heterodimer > IR-B = IGF1R / IR-B heterodimer [163]. However, some sources [15,49,156] imply that IGF-2 affinity to IGF2R is higher than to IGF1R.

### 6.1. IGF-1 Receptor

Earlier it was believed that insulin-like growth factor 1 receptor (IGF1R) mediates most, if not all, IGF-2 effects on cellular proliferation, survival, differentiation and migration [157,163]. However, the fact that IGF-1 also binds to IGF1R and in general, elicits the same effects with greater potency, led to the hypothesis about the specific purpose of IGF-2 that may be realized through some other receptors; this hypothesis was later confirmed [163,164,165].

IGF1R and IR are closely related members of the class II RTK family and these two receptors share high structure homology. Like other RTKs, both IGF1R and IR are transmembrane receptors with catalytic kinase domains on their intracellular *C*-termini. IR and IGF1R are disulfide-linked homodimers. Each monomeric subunit consists of α- and β-chains. Dimeric structure of these receptors is fundamental to the mechanisms of both ligand binding and tyrosine kinase activation, involving intra-molecular trans-phosphorylation of β-chains [8,153,154]. Both receptors are autophosphorylated upon binding with their respective ligands and intracellular signaling events initiated by this are highly similar for IR and IGF1R. These signaling cascades include phosphatidylinositol 3-kinase—Akt (protein kinase B)—forkhead box protein O (PI3K-Akt-FOXO) pathway and MAPK/ERK kinase—extracellular signal regulated kinases 1 and 2 (MEK-ERK) cascade. Both these pathways regulate cell survival, proliferation, differentiation, protein synthesis, glucose and lipid metabolism and many other important processes in the cell [153,166,167]. Many other second messengers mediate effects of IR and IGF1R activation. Among these messengers are the major substrates of these receptors—insulin receptor substrates 1 and 2 (IRS1/2) and Shc proteins—and many other molecules for which association with these receptors is less studied. Apparently, differences in these minor substrates of two receptors may be responsible for distinct effects of each receptor. Still, there are no signaling mechanisms that are specifically engaged by only one of these two receptors (see [153] for review).

Another important detail about IR and IGF1R is their ability to heterodimerize, leading to the formation of hybrid receptors. Affinity of insulin/IGF-1 hybrid receptor to insulin, IGF-1 and IGF-2 depends on which IR isoform is included within the heterodimer [163,168]. Still, little is known yet about these hybrid receptors specifically in the CNS.

IGF1R mostly mediates mitogenic effects [153]. It is crucial for the CNS embryonal development. Mice with homozygous brain-specific *Igf1r* knockout are microcephalic, have severe growth retardation and abnormal behavior [169]. Peak Igf1r mRNA expression throughout the brain is observed during the embryonal development [3,170]. The role of this receptor in the CNS development is mediated by PI3K-AKT signaling pathway that is involved in all stages of neuron formation and maturation: neural precursor cells (NPCs) proliferation [171], their survival (apoptosis prevention) [171,172,173] and differentiation into neurons [174]. According to some reports, MEK-ERK signaling cascade has similar functions in NPCs [175] but the data about its role in proliferation is controversial (see [176] for review).

After birth, IGF1R expression in the CNS drops and keeps decreasing later during adult life. However, IGF1R is still expressed at noticeable levels in neocortex, hippocampus, hypothalamus and brainstem of adult rodents. The highest IGF1R concentration in adult rat and mouse brains is observed in the choroid plexus, meninges and vascular sheaths [3,166]. Some mitogenic effects of IGF1R are still observed in adult brain, where IGF1R is necessary for many aspects of adult neurogenesis. Experiments with transgenic mice overexpressing IGF-1 specifically in NSCs demonstrated that IGF-1 upregulates the proliferation of NSCs by triggering MEK-ERK pathway signaling in the adult neurogenic niches, SVZ and subgranular zone of the dentate gyrus (SGZ). IGF-1 also induces differentiation of NSCs via the PI3K-Akt pathway [177]. We may suggest that IGF-2 may be able to trigger similar second messenger pathways in NSCs via IGF1R activation. The production of IGF-2 in the NSC (SOX2-positive) population in the dentate gyrus was shown to be high in adult mice [1], while IGF-1 level in the dentate gyrus decreases with age, as was shown in rat [178]. Experiments with cell cultures and local knockdown of IGF-2 expression in vivo demonstrated that locally produced IGF-2 acts via Akt signaling to regulate proliferation of NSCs in the dentate gyrus but not in the SVZ [1]. 

Overall IGF1R level in the brain decreases with age, just like IGF-1 level. Still, there is some data about IGF1R activity in adult brain regions, where IGF-1 usually works as a modulator of neuronal activity. For example, it was shown that presynaptic IGF1Rs in hippocampal neurons are basally active and necessary for adjusting glutamate release and the mechanism of this regulation involves changes in mitochondrial Ca^2+^ buffering [179]; it was also demonstrated that IGF-1 application increases excitability of these neurons via presynaptic mechanisms [180]. 

IGF1R in adult brain also provides regulation of energy metabolism; in astrocytes, IGF1R signaling is necessary for normal mitochondrial function. IGF1R-deficient astrocytes displayed altered mitochondrial structure and function, decreased mitochondrial oxygen consumption rate and increased mitochondrial reactive oxygen species (ROS) production. Astrocytic uptake of glucose is essential to supply the energy needs of neurons by releasing lactate that promotes hippocampal learning and memory; in astrocytes with IGF1R deficiency glucose uptake was reduced and lactate production (measured indirectly) was decreased [181]. Conversely, a study by Hernandez-Garzón et al. demonstrated that IGF1R inhibits activity of astrocytic glucose transporter GLUT-1 and restricts glucose uptake in somatosensory cortex [182].

The importance if IGF1R in brain activity in general and learning and memory formation processes in particular was demonstrated in multiple studies [181,183,184,185] but we could only find one research about the role of IGF-2/IGF1R interaction in memory [48]. 

We may conclude that currently published studies about IGF1R in the CNS demonstrate that its expression gradually decreases with age and its mitogenic function in adult brain persists only in neurogenic niches. Still, in other regions of adult brain this receptor participates in metabolic and homeostatic regulation. IGF1R activation and/or increased level of its cognate ligand IGF-1 were associated with different forms of neuronal activity and learning [48,183,185,186]. But some studies show that IGF-1 and/or IGF1R are not instrumental for memory formation. For example, a bilateral injection of IGF-1 into the dorsal hippocampus had no effect on hippocampal-dependent memory [187]. IGF-2 is much more often associated with memory enhancement than IGF-1 [12,165,187] but IGF-2 participation in learning and memory is primarily mediated by IGF2R [12,165]. While IGF-2 is capable of interaction with IGF1R, the role of this receptor in IGF-2-dependent processes in the brain is most likely not important, since direct inhibition of IGF1R did not affect IGF-2-mediated memory enhancement [11]. 

We may suggest that IGF1R activation associated with learning is more important for regulation of metabolism of activated neurons or for synaptic modulation but not for memory formation mechanisms per se. The fact that IGF-1 utilized by activated neurons is mostly imported from blood rather than produced locally [185] corroborates this. However, the role of IGF1R in memory formation may be more direct in neurogenic niches like the SGZ of the dentate gyrus that expresses IGF-2 in a relatively stable manner; the importance of IGF-2/IGF1R signaling in this zone for one particular kind of learning was already demonstrated [48]. 

There is another line of evidence that IGF1R function in adult brain may be redundant. Surprisingly, in mice heterozygous knockout of *Igf1r* gene in the brain or in all tissues increased the lifespan by 9% [169] or 26% [188], respectively. Another indirect proof of life-shortening effects of brain IGF1R signaling is that IGF1R level in the brain negatively correlates with longevity in different rodent species [189]. The negative role of IGF1R signaling in aging may be associated with toxic protein aggregation—in *C. elegans*, IGF1R-dependent activation of worm analog of PI3K-Akt-FOXO pathway was linked to Aβ aggregation [190]. We may suggest that natural age-related decrease of IGF1R expression in the brain may actually have adaptive value and that IGF1R signaling in adult brain is mostly redundant.

### 6.2. Insulin Receptor

Despite the most important function of IR being the regulation of glucose availability to non-neuronal tissues, this receptor has a distinct role in the CNS. IR is widely expressed in rat brain, with the highest expression levels in the olfactory bulb, hypothalamus, cerebral cortex, cerebellum and hippocampus [4]. Compared with IGF1R expression levels, IR expression prevails in the CA1 hippocampal field and hypothalamic arcuate nucleus [191]. IR receptors in the brain are more important in the postnatal life than in the embryonic development; neuron-specific knockout of IR in mice does not affect brain development or neuronal survival [192].

Structurally, IR is a class II receptor tyrosin kinase very similar to IGF1R described above [8,153,154]. There are two splice isoforms of this receptor, IR-A and IR-B. Exon 11 that encodes 12 amino acids at the *C*-terminus of the α-chain is absent in IR-A [193]. IR-A is able to bind IGF-2 with high affinity [157,158]. The ratio between mRNAs encoding IR-A and IR-B is regulated in a tissue-specific manner in both humans and rats. In human adult tissues there is a predominant expression of IR-B [193] (but it must be noted that both isoforms are expressed in the brain [194,195]); in adult rat, however, some tissues such as muscle, pancreas, brain, placenta and spleen express preferentially IR-A [193]. IR-A is also the predominant IR isoform in human fetal tissues and cancer cells. Interestingly, the relative abundance of the IR-A isoform in fetal tissues was higher than in corresponding adult tissues for all studied tissue samples except the brain [157].

Insulin, the principal ligand of IR, has 50% sequence homology with A and B domains of both IGFs. Apparently, insulin is not produced in the CNS in noticeable quantity [191]. Nevertheless, experiments in vitro show that neurons can produce insulin [196,197].

The CNS used to be considered an insulin-insensitive tissue. Glucose transporters GLUT-1 (in astrocytes), GLUT-3 (in neurons) and GLUT-5 (in microglia) account for the majority of glucose uptake in cells comprising CNS; all three of these transporters are not insulin-sensitive [191]. Recent studies challenge the notion of insulin being unimportant for glucose trafficking within CNS, showing that IR in the brain still participates in glucose metabolism regulation. IR signaling in astrocytes was demonstrated to control glucose-induced activation of hypothalamic proopiomelanocortin (POMC) neurons. These neurons are important for regulation of feeding behavior and whole-body glucose metabolism regulation. Astrocytic IR was also confirmed to be necessary for regulation of both glucose and insulin entry through the blood-brain barrier (BBB) [198].

Both IR-A and IR-B receptor isoforms are present in the brain. In humans, neurons almost exclusively express IR-A, while IR-B is predominantly expressed in astrocytes [194]; however, recently IR-B expression was detected in neurons from human brain samples [195]. Compared to peripheral tissues, IRs in rat brain have reduced N-glycosylation with sialic acid [4,199]. Experiments with mice showed that in peripheral tissues, like liver and muscle, IR ligand binding induces IR interaction with neuraminidase 1 that causes IR desialylation; vice versa, IR desialylation favors IR activation [4,200]. In the brain, IR is apparently not processed by neuraminidase 1, since rat brain IR was insensitive to treatment by bacterial neuraminidases in vitro [199].

There is some data about brain IR receptors having not only distinct structure but also some specific functions related to neuronal activity [4]. In rats, localized downregulation of this receptor in the hypothalamus leads to depressive-like behavior [201], while IR knockdown in the hippocampus causes impairments in spatial learning and impaired hippocampal neuroplasticity in brain slices, associated with AMPA and NMDA glutamate receptors dysfunction [202]. Endogenous changes in the brain insulin/IR system were documented as a result of learning in rats—after water maze training, IR expression in the CA1 and dentate gyrus of the hippocampus is upregulated [203] but down-regulated in the CA3 region [204]. Also, insulin-stimulated in vitro phosphorylation of IR was detected in the synaptic membrane fraction only from trained animals, suggesting that training may induce enhanced receptor sensitivity [203].

In murine SVZ, a site of active adult neurogenesis, IR-A (but not IGF1R or IGF2R) was shown to mediate stemness-promoting actions of IGF-2 in NSCs [205]. Interestingly, IR-A was the most abundant receptor in comparison to either IGF-1R or IR-B in cultured neural stem/progenitor cells and in the SVZ; as cultured cells became more lineage restricted, their IR-A level decreased [62,205]. So, the role of IR-A in mature neurons is most likely different from its role in NPCs.

It should be noted that IR-A activation by insulin and IGF-2 induces different patterns of gene expression [206]. In the culture of fibroblasts derived from mice with a targeted disruption of the *Igf1r* gene, ERK1/2 activation persisted longer after IGF-2 application, whereas Akt activation persisted longer after insulin application. In this study, the activity of p70S6 kinase (p70S6K) that plays a crucial role in cell proliferation was used to assess the activity of mitogenic pathways. Cells stimulated with IGF-2 had a higher p70S6K/Akt activation ratio than cells stimulated with insulin. Thus, some authors propose that while insulin/IR-A interaction mostly activates metabolic regulation pathways, IGF-2/IR-A interaction has more mitogenic effects [207]. Moreover, IGF-2/IR-A interaction was linked with carcinogenesis [4,208].

We may conclude that IR in the CNS participates in both neuronal function and glucose metabolism regulation. Some results indicate that insulin/IR signaling in the brain is different from other tissues [4] and that the role of this signaling pathway in the brain function including learning and memory is worth further study. IGF-2/IR-A signaling seems to be functionally different from insulin/IR signaling and more important for cell proliferation [206,207] and it also deserves more attention in the future research.

### 6.3. IGF-2 Receptor

IGF2R, also known as cation-independent mannose 6-phosphate receptor (CI-MPR), belongs to the P-type lectin family. This receptor is ubiquitously expressed [43]. IGF2R is able to bind IGF-2 with much higher affinity than IGF-1 and has no affinity to insulin [156,160,161]. IGF2R has a very large *N*-terminal extracytoplasmic region that binds the receptor ligands, a transmembrane region and a *C*-terminal cytoplasmic tail [15,155,156]. These receptors may exist as dimers [15]. Unlike IGF1R and IR, *C*-terminal of IGF2R does not have kinase activity and is mainly necessary for the receptor transport between different cell compartments (the Golgi apparatus, endosomes and cell surface). *N*-terminal extracytoplasmic region of IGF2R has a segmented structure, consisting of 15 homologous repeats with binding sites for various extra- and intracellular ligands. Among these ligands are molecules tagged with mannose-6-phosphate (m6p) (four m6p binding sites are located on segments 3, 5, 9, 15) and IGF-2 (its binding site is on the segment 11, with residues in domain 13 increasing the affinity of the interaction) [162].

Evolutionarily, the ability to bind IGF-2 was not the original function of the IGF2R molecule and emerged quite recently, first appearing in viviparous mammals [43], apparently as a countermeasure for IGF-2 excessive mitogenic effects [209,210]. IGF-2/IGF2R interaction supposedly leads to endocytosis, endosomal transport and eventual lysis of IGF-2 [211], consistent with the evolutionarily conserved role of IGF2R in the lysosome biogenesis [155]. 90% of IGF2R molecules are localized inside the cell, where they mediate transfer of about 50 different enzymes to lysosomes [155,212] after recognizing their m6p signal, which is added to these molecules during their transit through the ER-Golgi biosynthetic pathway [43]. IGF2R turnover between the trans-Golgi network (TGN), clathrin-coated vesicles and endosomes [156,213] depends on retromer cargoselective complex [214] and a number of other membrane trafficking proteins [215,216,217,218]. IGF2R internalization and retrograde trafficking from the cell surface to the Golgi are apparently mediated by additional adaptor system—clathrin adaptor proteins AP-1 and AP-2 [219,220].

About 10% of IGF2R molecules in the cell leave the endolysosomal cycle and get transported to the cell surface [43,156], where different *N*-terminal segments of the receptor bind various extra- and intracellular ligands, leading to their activation, degradation, endocytosis and/or trafficking. In addition to IGF-2, IGF2R could bind a number of other molecules: growth factors, cytokines, m6p-tagged proteolytic enzymes, transforming growth factor beta (TGF-β), urokinase-type plasminogen activator receptor (uPAR) and some others [15]. Upon binding with IGF2R, these molecules undergo endocytosis and then are subject to degradation in lysosomes (like in the case of IGF-2) [221,222], proteolytic activation [223] or intracellular trafficking [224,225]. Some data indicate that proportion of IGF2R molecules on the cell surface is dynamically regulated depending on the cooperation between different receptor ligands [226]. Internalization of IGF2R was found to be much faster after binding with a multivalent ligand (β-glucuronidase bearing multiple m6p moieties) compared with IGF-2 [227]. This may indicate that m6p-tagged enzymes and growth factors regulate the redistribution of IGF2R molecules by independent mechanisms.

Since IGF2R does not have kinase activity, its possible direct participation in signaling pathways is still questioned. Nevertheless, a growing body of evidence supports indirect IGF2R signaling mediated by G-proteins. Despite the structural difference of IGF2R from classic G-protein-coupled receptors (GPCRs), the association of IGF2R with G_s_ [228] and G_i_ proteins [229] (including separate Gα and Gβγ subunits) [229] that is stimulated by IGF-2 binding [230] was demonstrated in vitro in cell-free systems and in monkey fibroblast-like cells and in cardiac cells both in vitro and in vivo [231,232,233,234,235]. A number of studies also showed that IGF2R activation is linked with G-protein dependent effectors: ERK1/2 [45], protein kinase C (PKC) [46], sphingosine kinase (SK) [47,156]. Still, attempts to register direct binding between IGF2R and G-proteins in a cell-free system or in mouse fibroblast-like cells were not successful [236], which could suggest cell specificity and/or indirect nature of this interaction. 

IGF2R is widely expressed in most regions of rodent brain [237,238]. In the rat brain tissue, IGF2R is mainly expressed by neurons, with glial expression being either absent or very low [165]. Among different kinds of neurons, IGF2R expression is apparently enriched in some particular subtypes, like excitatory pyramidal neurons and granule cells of the hippocampus [238] or neurons expressing a cholinergic phenotype [239]. Despite low baseline IGF2R level in glial cells, its glial expression may be induced in some pathological conditions: in mouse Schwann cell-like cells it was reported to happen as a response to demyelination [240], while in human microglial cells, IGF2R expression induction could be a part of their response to the proinflammatory cytokine IFNγ [241]. Reports about subcellular neuronal localization of the receptor are controversial: in early studies, IGF2R immunoreactivity was mainly reported in dendrites and axon terminals [239] but most recent works describe it as being predominantly somatic [165,242]. In rat hippocampus, IGF2R is mostly expressed in the pyramidal cell layer of the CA1-CA3 subfields and in the granular cell layer of the dentate gyrus, whereas IGF1R density is higher in the molecular layer of the dentate gyrus and the CA2-CA3 hippocampal subfields [161]; the two receptors not having the same distribution may suggest that IGF2R may have some other functions beside scavenging the excess IGF-2.

First evidences of IGF2R having not only homeostatic but also signaling functions in the CNS were acquired indirectly, after examining the effect of IGF-2 on the release of neuropeptide Y (NPY) ex vivo. In an experiment with application of insulin, IGF-1 and IGF-2 to microdissected rat brain sites, it was shown that both insulin and IGF-2 (but not IGF-1) decrease the release of NPY in preparations of paraventricular nucleus [243] and an IGF2R-dependent mechanism of this effect was suggested. In the following research with ex vivo hippocampal slices, IGF-2 was found to stimulate acetylcholine release (while IGF-1 inhibited it) and both peptides were then confirmed to bind with their own receptors [161].

Further studies allowed more direct approach to study IGF2R receptor, made possible by development of its artificial selective agonist, Leu^27^IGF-2; experiments with this molecule confirmed that acetylcholine release stimulation observed earlier after IGF-2 application to hippocampal slices is indeed mediated by IGF2R, excluding other receptors [244]. Another research using the same artificial agonist showed that IGF2R activation leads to inhibition of GABAergic parvalbumin-reactive neurons in acute slices of rat hippocampus and cortex (but not in the striatum slices) [245]. Both these works proposed a G-protein mediated mechanism of IGF2R activation (specifically, G_i/o_-PKCα signaling pathway), based on the sensitivity of observed effects to the G_i/o_-protein inhibitor pertussis toxin; this conclusion was further supported by a later study where inhibitors of GPCR/G-protein coupling were used [246]. A possible connection between IGF2R activity and G-protein effectors ERK1/2 in neurons was also reported by Schmeisser et al. [98]; in this work IGF2R—ERK1/2 signaling was linked to dendritic spine maturation and increased expression of GluA1 (AMPA receptor subunit) and postsynaptic scaffold proteins PSD95, SAP97 in mouse forebrain neurons. Still, it should be noted that all the links observed here between IGF2R, G-proteins and their effectors were indirect.

The experiments performed by Schmeisser et al. also addressed the role of native IGF2R ligands (and endogenous IGF-2 in particular) in mediating the receptor activity. Inactivation of NF-κB signaling pathway (that was shown by the authors to be necessary for *Igf2* expression) in mutant mice led to the reduction of Igf2 mRNA by 80%–90% in hippocampal cultures and provoked synaptic deficits that were completely reconstituted by incubating cultures with medium from control cultures for 24 h. Since the same result was achieved by the application of recombinant IGF-2, the authors explained the effect by the autocrine/paracrine secretion of IGF-2 into the control medium. It should be noted, however, that the time frame for the endogenous IGF-2 synthesis in this study was not specified [98].

IGF2R was recently found to mediate IGF-2 function as a memory enhancer, even while the exact mechanisms of this are still unclear. The studies demonstrating the role of IGF-2/IGF2R interaction in learning and memory are described in more detail in the section “IGF-2 as a memory enhancer” but here we provide a short summary. In all the studies listed below, the role of IGF2R in the observed effects was confirmed directly. Chen et al. [11] showed that hippocampal injection of IGF-2 enhances memory retention in rats. Similar observations were made by Lee et al. [40]. Stern et al. [12] used systemic administration of IGF-2 and demonstrated an enhancement of learning and an increased expression of proteins encoded by immediate early genes (IEG) in mice. The very recent study by Yu et al. [165] confirmed some of the previously described effects and also revealed that despite IGF2R activation being crucial shortly after training, the levels of IGF2R mRNA and protein remain unchanged, suggesting that IGF2R trafficking and redistribution is necessary for these mechanisms. Importantly, in the same study it was demonstrated that another IGF2R ligand—m6p—when injected in the same manner as IGF-2 (in the hippocampus in rat or systemically in mouse), has the same effects as IGF-2. IGF-2 and m6p were also found to be interchangeable for IGF2R-mediated effects in reversing the cognitive impairment in the model of Angelman syndrome [247].

Thus, IGF2R is largely involved in the protein turnover in general, participating in protein trafficking, lysosomal proteolysis and regulation of de novo translation of some proteins like IEG products. The important role of IGF2R in protein metabolism in neurons was also shown in a number of works in the field of neuropathology. For example, the lysosomal function of IGF2R may apparently be induced to compensate for overall protein hyperproduction in a mouse model of autism. In BTBR *T^+^ Itpr3^tf^*/J mice that display a behavioral phenotype similar to that of autism-spectrum disorders, IGF-2 injection reversed the cellular abnormalities associated with protein overproduction. These effects of IGF-2 were mediated by IGF2R (but not IGF1R) [111]. IGF-2 was also shown to reduce the accumulation of intracellular polyglutamine proteins in Neuro2a cells and of mutant huntingtin in neuronal cultures derived from Huntington’s disease (HD) patients. The decrease in protein aggregation was independent of the activity of the proteasome or autophagy pathways. Instead, IGF2 treatment enhanced the non-conventional disposal of soluble polyglutamine peptide species into the extracellular space through the secretion via microvesicles and exosomes. Knockdown of IGF2R decreased almost in half the levels of polyQ_79_-EGFP secretion, whereas the knockdown of IR or IGF1R did not have any effect on this [49]. Similar results were obtained in a study using a genetic model of AD: in the primary hippocampal cultures from Tg2576 mice overexpressing amyloid-β-precursor protein (APP), transduction with an IGF-2-expressing vector caused a significant reduction of the extracellular Aβ_42_ level measured in the culture medium. The importance of IGF2R in this mechanism was confirmed by anti-IGF2R antibody partially blocking Aβ_42_ clearance in the control cultures conditioned with Aβ_42_-containing medium from the Tg2576 cultures [20]. The mechanism of IGF2R stimulation improving AD-associated neuropathologies is likely connected with the role of this receptor in cholinergic neurons. The importance of IGF2R expression in cholinergic neurons was repeatedly demonstrated [239,248,249] and some reports suggest that IGF2R expression in these neurons may increase to compensate some dysfunctions. For example, in an experiment with 192 IgG-saporin-induced loss of rat basal forebrain cholinergic neurons, increased IGF2R expression was found to be associated with surviving neurons [250].

Considering the role of IGF2R in lysosomal function, cellular catabolism and autophagy, IGF2R signaling may supposedly be adjusted to compensate for various protein metabolism abnormalities [111]. On the flip side, disruptions of this signaling may trigger various lysosomal deficits that were noted in neurodegenerative diseases like AD and Parkinson’s disease (PD) [251,252,253], as well as in multisystem genetic disorders involving intellectual disability (Lowe syndrome) [254]. It was shown that in AD, lysosomal system pathology is associated with disrupted retromer-mediated endosome-to-Golgi retrieval of IGF2R [255]. It should be noted that both decreased and increased activity of IGF2R in the TGN may lead to pathologies. For example, while in aforementioned genetic models of AD [20,256] or after 192 IgG-saporin injection [250] (a common pharmacological model of AD cholinergic deficits [257,258]) IGF-2-dependent IGF2R stimulation or endogenous IGF2R expression increase were associated with neuroprotection, there are some data indicating that an excess of IGF2R may be harmful for the cell and may even lead to Aβ accumulation. When IGF2R was artificially overexpressed in mouse fibroblasts, it led to increased levels of APP and β-site APP-cleaving enzyme 1 (BACE1), resulting in enhanced Aβ production. Moreover, cells overexpressing IGF2R were found to be more vulnerable to a cytotoxic agent [259]. 

These contradictions may be explained by the existence of different molecular pathways in which IGF2R participates with and without binding IGF-2. Also, pathological effects of IGF2R overexpression suggest that in healthy cells, some mechanisms that rigorously control IGF2R level must take place. Further research is necessary to reveal the exact nature of these mechanisms.

Taken together, the described studies demonstrate the diversity of IGF2R neuronal functions, indicating its multifaceted role in both lysosome biogenesis [155,156,212,213] and neuronal activity [98,161,243,244,245,246]. Such complexity of functions could explain the importance in IGF2R for different forms of learning and memory demonstrated in vivo [11,12,13,165,187], especially considering the important, if largely overlooked, role of the endosomal system in governing the spatio-temporal distribution of signaling molecules [260]. Still, the actual molecular mechanisms mediating IGF2R signaling and likely involving either new protein synthesis or activity-dependent protein degradation [261,262,263] that may possibly be linked with the lysosomal function of IGF2R, require further research.

Recently, it was shown that another endogenous ligand of IGF2R, m6p, may induce effects identical to those of IGF-2 [165,247]. M6p tag is attached to many lysosomal enzymes and there is some data that these enzymes may undergo Ca^2+^-dependent exocytosis in rat hippocampal pyramidal neurons [264]. Thus, it is possible to assume that IGF2R activation could be involved in a positive feedback loop, depending only on Ca^2+^-dependent release of m6p-tagged molecules, which the same receptor previously carried. That way, receptor activation would not require any additional ligands (like IGF-2). Nevertheless, IGF2R activation may also involve interaction with m6p-tagged proteins originating from other cells—in this context, it is important to note that the overall brain glycoproteome is assumed to contain 2-8-fold more m6p-tagged proteins compared with other tissues [265] and among these proteins are m6p-tagged receptors and functional partners of IGF2R, such as GABA_B_ [266].

Downstream signaling pathways of IGF2R activation are even less understood; in particular, very little is known about how this receptor affects protein synthesis and especially the synthesis of IEG products. 

Despite IGF2R lysosomal function being interconnected with protein degradation, some studies show the role of IGF2R in protein synthesis as well, with IGF2R inhibition in neurons preventing the learning-induced increase of overall translation level [165]. Possible link between lysosomal-endosomal system and translation in neurons was previously suggested in the work by Cioni et al., performed on *Xenopus* retinal ganglion cells [267], where axonally transported late endosomes were shown to function as mRNA translation platforms. Given that IGF2R participates in protein trafficking to late endosomes/lysosomes [155,212] and that activity-dependent transport of lysosomes was already demonstrated in dendrites [263], it is quite possible that this receptor may regulate endosomal availability and trafficking required for protein synthesis, as was suggested in [165]. Lyso/endosomes may also carry amino acids, which could facilitate protein synthesis by providing the necessary substrate.

Involvement of IGF2R activity in memory formation could depend on a number of others, possibly indirect mechanisms: for instance, “trans-activation” of different G-protein associated receptors, as was already proposed for cholinergic [244] and GABAergic systems [245]. Another intriguing possibility is the extracellular matrix remodeling that was proposed as a mechanism for long-term memory storage [268] and is largely dependent on matrix metalloproteinases (MMPs) [264]. Indeed, MMPs require for their activation two molecules that associate with IGF2R—plasmin [269,270,271] and cathepsin B [264,272,273]. Plasmin was previously shown to be activated by cleavage on the cell surface via IGF2R-dependent mechanism [274], while cathepsin B is transported to lysosomes with the help of m6p receptor system [275] and is subject to activity-dependent exocytosis [264], as described above. Proof-of-principle for MMP-9 activation involving IGF-2 and IGF2R with subsequent matrix remodeling was provided by experiments on cardiomyoblast cultures [276]; such mechanism could be one of the possible answers for the long-term-specific IGF2R-dependent effects. 

It is also of note that much of the discovered IGF2R-mediated effects are not indicative of a distinct memory stage—for example, IEG products, synthesis of which was shown to be dependent on IGF2R, characterize neuronal activity as a whole, whereas receptor knockdown only affected memory consolidation but not the ability to learn [165]. Thus, for further research, it is important to discover how IGF2R activity mediates memory consolidation processes, in comparison to processes involved in learning.

In summary, IGF2R has a conserved role in lysosome biogenesis, is also necessary for trafficking, activation or degradation of multiple extracellular substances and likely also has a signaling activity but its mechanisms are still unclear. The ability of this receptor to bind a variety of both intra- and extracellular ligands likely places it in the intersection of different signaling pathways. Thus, it is possible that IGF2R does not have its own distinct downstream signaling cascades but functions as a fine-tuning modulator for signaling of other receptors. There are also some interesting possibilities of IGF2R involvement in neuronal function being connected with the role of this receptor in lysosome trafficking and/or exocytosis of lysosomal enzymes. More research is necessary to establish the mechanisms of IGF2R signaling in neurons.

## 7. IGF-2 as a Memory Enhancer

The role of IGF-2 in learning and memory was discovered quite recently; nevertheless, there are already multiple studies confirming that IGF-2 may function as a memory enhancer [11,12,40,165]. This effect of IGF-2 is currently considered to be mediated mostly via IGF2R [11,40,165] but in one case the role or IGF-2 interaction with IGF1R was shown to also be important for memory-related processes [48].

So far it was established that IGF2R activation (1) is necessary for long-term hippocampus-dependent memory formation [13,165] but not for learning, for memory retrieval or reconsolidation or for short-term hippocampus-dependent memory [165]; (2) is connected with de novo protein synthesis and synthesis of IEG-encoded proteins in particular [11,165]; (3) must happen relatively early, in the first few hours after learning [11,40].

The first in vivo demonstration of IGF2R involvement in the memory formation was made in 2011 by Chen et al. [11]. The authors showed that in rats, hippocampal injection of IGF-2 immediately after passive avoidance training significantly enhances memory retention tested 3 weeks later, while anti-IGF2R antibody (but not a selective IGF1R inhibitor JB1) co-injected with IGF-2 completely abolishes the memory enhancement effect. Double intrahippocampal injections of the antibody alone (immediately and 8 h after training) caused a complete amnesia 24 h after training. The main effects of IGF2R activation were activation of synaptic glycogen synthase kinase 3 beta (GSK3β) and expression of GluA1 AMPA receptor subunit, both effects being closely linked to synaptic plasticity. IGF-2-mediated memory enhancement required de novo protein synthesis and Arc translation but did not depend on the transcription factor C/EBPβ, further suggesting that it may use local synaptic mechanisms, rather than cell-wide ones.

It should be noted that despite the importance of both IGF-2 and IGF2R for memory formation established in this study, not all details of their interactions were clarified. Large discrepancy between the time points of Igf2 mRNA transcription increase (20 h after training) and IGF2R involvement in memory formation processes (0–8 h) likely requires IGF2R activation to be transcription-independent. At least three different scenarios may be used for IGF2R activation before the newly produced IGF-2 is present: activation by pre-existing IGF-2 (for example, IGF-2 molecules released from IGFBPs in the extracellular space), rapid de novo IGF-2 translation from its mRNA that was locally “conserved” in the synapse or even IGF2R activation by some completely different ligand. The complicated nature of IGF-2/IGF2R interactions in memory enhancement is additionally confirmed by the fact that either IGF-2 or IGF2R inhibition only disrupted the memory if inhibitors were injected twice, at specific time points.

The role of endogenous IGF-2/IGF2R signaling in the memory formation was further confirmed in a number of following studies, extending IGF2R activity to different learning paradigms and time points. For example, a research by Stern et al. that used systemic (subcutaneous) administration of IGF-2 before training in mice [12] demonstrated significant enhancement of learning in different experiments, including fear conditioning, novel object recognition and Y-maze, with no adverse effects of an acute injection, as was demonstrated by a battery of physical, behavioral and sensorimotor tests. The observed effects of IGF-2 treatment were apparently mediated by hippocampal IGF2R and linked to the significant increase of the expression of proteins encoded by IEGs, *Arc* and *Zif268*, 1 h after injection. Another study, performed by Lee et al., examined IGF2R-dependent memory enhancement in the passive avoidance task caused by IGF-2 administration first reported by Chen et al. [11] but the effects were measured in entirely different time frames. The improved performance in the task (increased latency of entering the dark compartment measured at 24 h after training) was observed as a result of a bilateral injection of IGF-2 into the dorsal hippocampus at 6 or 9 h (but not 12 h) after training. Intriguingly, injections performed at any of the three aforementioned time points significantly improved the performance in the retention trial at 21 days after training [40].

Lastly, the role of IGF2R in memory formation was addressed again in a recent study [165], which confirmed some of the previously described effects: (1) disruption of memory consolidation after the IGF2R blockade, with double injection of its inhibitor being necessary for this effect and (2) the influence of IGF2R activation on translation (but not transcription) of IEG products.

The role of IGF1R in IGF-2 dependent memory processes was shown in a study by Agis-Balboa et al. [48]. These authors studied not memory formation but memory extinction. Using contextual fear conditioning paradigm, they demonstrated that the peak Igf2 mRNA expression in mouse hippocampus happens at 3 h after the reminder (the first extinction trial). Injection of IGF1R inhibitor immediately after each extinction trial impaired fear extinction, while IGF2R inhibitor injected in a similar manner had no effect (see Supplementary Figure S11 in [48]). As for the memory enhancement effect of IGF-2, IGF1R is most likely is not involved in its mechanism, since direct inhibition of IGF1R did not affect IGF-2-mediated memory enhancement shown in [11].

## 8. IGF-2 as a Neuroprotective Agent

Multiple in vivo studies demonstrated an increase of the level of IGFs or their receptors (measured as either mRNA or protein expression) within hours or days after the CNS hypoxia/ischemia [277,278,279,280]. These studies suggested that IGFs may affect neuron survival after a damaging influence, so the effects of these factors were studied in vitro and in vivo.

In an in vitro study, it was shown that application of IGF-1 or IGF-2 to rat embryonic neuronal culture prevents cell death caused by glucose deprivation [281] and that for both factors this effect directly correlates with concentration (in the range from 0.3 to 300 ng/mL). Interestingly, IGFs application also caused a significant decrease in intraneuronal Ca^2+^ concentration (that usually increases as a result of glucose deprivation). In the later work by the same authors, it was shown that IGF-2 application also normalizes mitochondrial transmembrane potential but does not prevent ATP depletion in cultured neurons deprived of glucose [282]. A few other works performed on neuronal cultures demonstrated that IGF-2 may improve redox status and regulate the expression of synaptic proteins in cultures after an oxidative insult. Application of corticosterone (10 µM) caused an increased ROS production in cortical cultures from adult rats; this was prevented by addition of IGF-2 (100 ng/mL). Moreover, IGF-2 application increased the expression of antioxidant enzymes and cytochrome *c* oxidase activity and partially normalized mitochondrial membrane potential disrupted by corticosterone toxicity. This antioxidant effect of IGF-2 was blocked by simultaneous inhibition of IGF1R and IGF2R but not by inhibition of any one of these two receptors. It should be noted that higher doses of IGF-2 (1000 ng/mL and 10000 ng/mL) did not have neuroprotective effects in corticosterone-treated cultures [42]. In another work, the same authors showed that corticosterone application leads to decreased expression of synaptic proteins like synaptophysin and PSD-95, as well as decreased synaptic function measured by FM 1-43 fluorescence intensity. IGF-2 (100 ng/mL) reversed these negative effects [2]. Additionally, exogenous IGF-2 restored synaptic density and promoted spine maturation in mutant IKK/NF-κB signaling-deficient, IGF-2-deficient mouse neurons. This result also confirms the ability of IGF-2 to improve neuronal function. In this work, application of recombinant IGF-2 or incubation in medium from control cultures led to similar results, so the authors concluded that control neurons secreted IGF-2 into the medium and that the observed action of IGF-2 normally involves its autocrine/paracrine secretion. On the other hand, IGF-1 application exhibited only a moderate effect on synaptic density compared with IGF-2 application [98].

In the studies in vivo, antioxidant effects of IGF-1 [283] and IGF-2 [284] were assessed in aging rats. Compared with young controls, untreated old rats showed a decrease of serum total antioxidant status; IGF-2 or IGF-1 injection restored total antioxidant capability in the serum of old rats to the level of young controls. IGFs injection also affected the levels of some but not all antioxidant enzymes in the cortex and hippocampus of old animals, making these levels more similar to those observed in young rats. Thus, these two papers demonstrate that both IGFs function as antioxidants. It should be noted that IGF-1 and IGF-2 effects were not identical in these experiments: for instance, IGF-1 increased free testosterone level in the serum of aging rats even higher than the level observed in young controls, while IGF-2 did not affect the testosterone level.

There is some controversy about the role of IGF-2 in neuroprotection during ischemic brain injury. For example, Mackay et al. [285] showed that intracerebroventricular (i.c.v.) injection of IGF-2 or IGF-1 reduces the volume of ischemic damage after a permanent occlusion of the middle cerebral artery. It must be noted that IGF-2 was as effective as IGF-1 at reducing the amount of brain injury but had a lower effective dose (5 µg) than IGF-1 (50 µg). This result is quite intriguing because IGF-1 has a higher affinity to IGF1R, so the lower effective dose of IGF-2 in this experiment may indicate that the neuroprotective effect of IGF-2 is mediated by more than one receptor. Interestingly, in the same study it was shown that IGFBP ligand inhibitors that displace endogenous IGFs from complexes with IGFBPs also reduce the volume of ischemic damage when injected i.c.v. But in another work [286] it was demonstrated that IGF-1 (30 µg i.c.v.) reduced the neuronal loss in different brain regions after hypoxic-ischemic injury in rats, while IGF-2 in the same dose either did not affect the neuronal loss or even increased it in some regions. IGF-2 also reduced the neuroprotective effect of IGF-1 if injected simultaneously. It is unclear why the effect of IGF-2 was different in these two papers but it should be noted that in the experiments performed by Guan et al. [286] ischemic damage induction included 10-min exposure to hypoxic atmosphere (6% O_2_). We may conclude that IGF-1 was found to have neuroprotective effect in different models of ischemic damage, while IGF-2 effects depend on experimental conditions.

In a model of hemorrhagic brain damage in rats, autologous blood injection to the hippocampus caused reduction of the hippocampus size and decreased the number of neurons in CA1 field. IGF-2 injection to the hematoma 30 min after its induction reversed these effects. Animals that had IGF-2 injected to the hematoma showed decreased infarct volume, increased size of the hippocampus and increased number of CA1 neurons compared with the group with untreated hematomas [287].

Endogenous IGF-1 was also shown to promote neuronal plasticity, neurogenesis and neuroprotection in the context of exercise. It was demonstrated that participation in physical activity delays onset of and reduces risk for AD, HD and PD and can even slow functional decline after neurodegeneration has begun (see [288] for review). IGF-1 is one of the principal growth factors that mediate the effect of exercise on the brain [288,289,290,291,292]; it performs this function together with BDNF and vascular endothelial-derived growth factor (VEGF) [288]. Blocking IGF-1 by systemic injection of anti-IGF-1 antiserum inhibited the survival-promoting effect of exercise on newly generated neural precursors in rats [289]. It should be noted that possible involvement of IGF-2 in exercise-mediated neuroprotection was not studied in these works. However, IGF-2 may also promote survival of nascent neurons and these effects are likely mediated via IGF1R. Agis-Balboa et al. showed that (1) fear extinction promotes the survival of newborn neurons (but not neurogenesis); (2) IGF-2 is necessary for both fear extinction and survival of newborn neurons; and (3) IGF-2 effect on fear extinction is mediated by IGF1R and not IGF2R, as was shown in behavioral experiments with specific inhibitors of both receptors [48]. All three results, taken together, allow us to suggest that IGF-2-dependent survival of newborn neurons is mediated by IGF1R, even while there is no direct confirmation of this in this study. Also, it was shown that both IGFs participate in hippocampal neurogenesis regulation [1,48,290]. A mutation in *Dgcr8*, a candidate gene for 22q11.2 deletion-associated schizophrenia, leads to decreased adult hippocampal neurogenesis in mice. It was shown that *Dgcr8* mutation also causes decreased mRNA and protein expression of IGF-2 in the hippocampus. Exogenous IGF-2 rescued the proliferation of mutant adult neural stem cells both in vitro and in vivo. Furthermore, IGF2 improved the hippocampus-dependent spatial working memory deficits in *Dgcr8*^+/−^ mice [26].

In the Angelman syndrome mouse model, Ube3^am-/p+^, systemic administration of IGF-2 significantly attenuated acoustically induced seizures and restored cognitive impairments assessed by measurements of contextual and recognition memories, motor deficits and working memory [247]. In a mouse model of autism spectrum disorders (ASD), BTBR *T^+^ Itpr3^tf^*/J strain, IGF-2 injection reversed the cellular abnormalities associated with protein overproduction: activation of the AMPK-mTOR-S6K pathway, increased translation in synapses and decreased autophagy level [111].

Neuroprotective properties of both IGFs were also demonstrated in a study that used APP transgenic mice Tg2576, a genetic model of AD. Some of these effects including Aβ clearance were confirmed to be mediated by IGF2R as discussed above. In behavioral tests, it was shown that ectopic hippocampal expression of IGF-1 or IGF-2 increases contextual freezing in 16 months old Tg2576 mice. However, only IGF-2 was able to improve the learning in both contextual fear conditioning and Morris water maze test in 20 months old mice [20]. In another genetic model of AD, APPswe.PS1dE9 mice, i.c.v. administration of IGF-2 not only reduced the number of hippocampal Aβ-positive plaques but also partially restored the damages in cholinergic system: IGF-2 treatment increased the levels of both choline acetyltransferase (CHAT) and cholinergic differentiating factor, BMP9 [256].

Another neurodegenerative disorder, HD, may be modeled in vitro using medium spiny neurons derived from induced pluripotent stem cells (iPSC) generated from patients’ fibroblasts. It was shown that IGF-2 overexpression reduces the level of mutant huntingtin in iPSC-derived HD neurons, while in neurons derived from iPSCs of control subjects IGF-2 did not affect the expression of wild-type huntingtin. Experiments with Neuro2a cells demonstrated that IGF-2 overexpression also reduces the accumulation of intracellular polyglutamine proteins and that this effect is mediated by IGF2R but not IGF1R or IR. Moreover, IGF-1 overexpression did not cause the same effect. In the same study, administration of IGF-2 into the brain of HD mice using gene therapy led to a significant decrease in the levels of mutant huntingtin in three different animal models [49]. Protective action of IGF-2 was also shown in human motor neurons derived from iPSCs generated from patients with spinal muscular atrophy (SMA). These neurons present an apparent cell autonomous degeneration in vitro, which is evident after 8 weeks of culturing. IGF-2 application to the culture at 4 weeks with subsequent maintenance led to motor neurons protection and improved their survival at 8 weeks of culture. Also, morphometric analysis showed that IGF-2-treated SMA neurons have increased neurite lengths (compared with untreated SMA neurons). In the same study, iPSCs were used to model another neurodegenerative disease, amyotrophic lateral sclerosis (ALS). Motor neurons derived from ALS patient iPSCs display some signs of the disease in vitro but cell degeneration in such cultures is not evident. To model ALS in vitro with these cells, two different assays were used: glutamate excitotoxicity or co-culture with astrocytes from SOD1^G93A^ mice (a genetic model of ALS). Pretreatment with IGF-2 protected motor neurons in both ALS-like toxicity assays. In an animal model of ALS, viral delivery of IGF-2 to motor neurons of 80 days old SOD1^G93A^ mice displaying extensive muscle denervation prolonged their life-span by 14 days [293]. It should be noted that in a similar experiment, viral delivery of IGF-1 to motor neurons of 90 days old SOD1^G93A^ mice extended their median life-span by 22 days [294].

Thus, both IGF-2 and IGF-1 have neuroprotective effects in different experimental models. Both IGFs had similar effects in a model of glucose deprivation-induced cell damage [281,282], as well as in one model of ischemic brain injury [285]. Nevertheless, in a model of hypoxic-ischemic brain damage [286] IGF-1 had neuroprotective effects while IGF-2 increased neuronal loss. Both IGFs had a positive effect on redox status of the serum and the brain of aged rats [283,284] and protected motor neurons in SOD1^G93A^ mice [293,294]. In some other neurodegenerative disorder models, neuroprotective effect of IGF-2 is more prominent compared with IGF-1 effect, especially when cognitive functions or protein aggregation are considered [20,49]. Apparently more noticeable effect of IGF-2 on preventing accumulation of toxic proteins like Aβ or mutant huntingtin is connected with IGF2R activation. This receptor binds IGF-2 with much higher affinity than IGF-1 and its function is linked with lysosomal system activation [156].

## 9. Discussion

Experiments on rodents with artificial increase of systemic or local IGF-2 concentration, that was achieved either by an injection of the recombinant protein or by ectopic gene expression, demonstrated that IGF-2 may alleviate cognitive deficits in animal models of AD [20], ASD [111], Angelman syndrome [247], Fragile X syndrome [101], as well as aging-related cognitive decline [108].

Neuroprotective properties of IGF-2 were confirmed in neuronal culture experiments [2,42]; one important example of this is the ability of IGF-2 to rescue SMA motor neurons from degeneration [293]. Moreover, IGF-2 application reduced protein aggregation in models of HD [49] and transduction with IGF-2-expressing vector caused a significant reduction in extracellular Aβ_42_ level measured in the culture medium from Tg2576 cultures [20]. All these data suggest that IGF-2 system is a promising target for developing drugs that improve the CNS function.

Growth factor-based pharmaceuticals were proposed to be used in clinical practice for regeneration of various tissues [295]. But the complications that arise in studying growth factors in general have to be considered. We will discuss some of these complications here.

One of the challenges is translation of animal test results to humans. Some physiological traits that are specific to humans may lead to various substances having different effects in humans compared to model organisms. Because of this, for translational studies it is important to understand the species-specific differences in the function of endogenous growth factors.

In the present review we described such differences in IGF-2 system between rodents and humans. The most important difference is *IGF2/Igf2* gene expression level in adults. While in adult rodents the highest *Igf2* expression is observed in the choroid plexus and the meninges and the expression of this gene in the liver is very low [28,59], in adult humans *IGF2* is expressed in various organs, including liver, adipose tissue and some others, at a relatively high level (see Figure 3 in [33]). Moreover, different IGF-2 proteoforms are present in the serum of rats and humans: while adult rats mostly have pro-IGF-2 in their serum, in adult human serum mature IGF-2 prevails and “big IGF-2” and pro-IGF-2 forms are present in much less quantities [54]. These species-specific differences in *IGF2/Igf2* expression may be explained by differences in postnatal development: a few day old rat pups have high Igf2 mRNA level in their liver (see Figure 7 in [55]) and different IGF-2 proteoforms, including mature IGF-2, in their serum [54]. One of the possible explanations of the high *IGF2* expression level in adult humans is the fact that the human gene has an additional promoter that is absent in rat and mouse orthologs [29].

Despite highly differing expression of *IGF2/Igf2* in the peripheral organs of humans and rodents, it seems that brain expression patterns of this gene are more similar between species. More specifically, in both human and rat brain endogenous *IGF2/Igf2* expression is relatively low (except in the choroid plexus and the meninges) [33,59,85,86]. The available information about *IGF2* expression in the human choroid plexus is quite sparse but the analysis of bioinformatics data shows that *IGF2* is one of the highly expressed genes in this structure, as compared to other genes [85,86]. Moreover, in humans *IGF2* expression in the neuronal tissue proper is not very high, which also concurs with the data for rodents showing that *Igf2* expression in the rat choroid plexus is much higher compared to that in the hippocampus [28]. 

We may suggest that wild-type rats and mice should be used very carefully to study the effects of IGF-2 and its analogs in most cases where any kind of conclusions about the effect of the same pharmaceuticals in humans is expected. For example, in the papers by Steinmetz et al. [111] and Yu et al. [165], cognitive enhancement effects were observed in mice after systemic injection of recombinant mature IGF-2. This protein is not normally present in the adult rodent serum, while in human serum mature IGF-2 level is quite high (even if most of it is bound with IGFBPs [119,120]). Thus, it is unclear whether the effect of such injection may be reproduced in humans that have very different baseline serum level of endogenous mature IGF-2 form.

We may probably only translate some results from wild-type rodents to humans in cases where the aspects of IGF-2 system important for a study are similar between species. In a work by Pascual-Lucas et al. [20], cognitive improvement effects were demonstrated after intrahippocampal injection of a viral vector expressing *Igf2*. We may suggest that effects of localized increase of *IGF2* expression in the human brain may be similar, because like in the rat brain, endogenous *IGF2* expression in the human brain is quite low [33,59].

Interestingly, in some other species, like cows, sheep and some primates, *IGF2* expression profiles are more closely resembling the human one than the rodent one. We may suggest that instead of wild-type rodents, translational research concerning IGF-2 system should be performed either on other species that are more human-like in this aspect or on specially created genetically modified rodent strains with human-like *Igf2* expression pattern. We hope that some future studies will provide a suitable animal model for studying IGF-2. Alternatively, a novel and very promising approach for modeling human neuropathologies in experiments is to use induced pluripotent stem cells (iPSCs). Somatic cells are taken from patients and reprogrammed to pluripotency in vitro; after this, they may be artificially differentiated into most types of cells including neurons [296] and may even be used to generate brain organoids [296,297]. iPSCs have been already successfully used in studying IGF-2 effects in neurodegenerative disorders like HD [49], ALS and SMA [293].

Another important aspect of designing drugs based on growth factors in general is that growth factors may trigger malignant transformation [163,298]. Many growth factor receptors are involved in cell proliferation regulation [299]. In our review we discussed three receptors that are able to bind IGF-2: IR (primarily IR-A isoform), IGF1R and IGF2R. IR-A and IGF1R are “classic” growth factor receptors that belong to the RTK superfamily. The role of RTKs in general in carcinogenesis is well established [300]. On the other hand, IGF2R is not an RTK and has distinct structure and activation mechanisms.

Is it possible that IR-A or IGF1R activation may have deleterious consequences because it triggers proliferation? On the one hand, there is some data about these receptors being involved in carcinogenesis, so strategies for treating some tumors based on inhibition of IGF1R and IR-A are being developed [299,301,302,303]. On the other hand, promising medications for treating various diseases were designed based on IGF-1 (another IGF1R ligand) and these substances are already under evaluation in clinical studies [304,305,306]. As for IR, its cognate ligand is insulin, a thoroughly studied molecule that is widely used in medical practice and its benefits outweigh potential side effects. But it should be noted that the effects of IR-A activation by insulin and IGF-2 are not the same. There are some studies showing that these two ligands trigger distinct signaling cascades and while insulin/IR-A interaction mostly activates metabolic regulation pathways, IGF-2/IR-A interaction has more mitogenic effects [207]. Moreover, IGF-2/IR-A signaling may promote carcinogenesis, especially in conditions with increased IR-A/IR-B ratio (like obesity and type 2 diabetes) [4,208]. Thus, potential IGF-2 mimetic agents should be used very carefully, considering IGF-2 effects on cell proliferation.

It was shown in a number of studies [11,111] that memory-enhancing action of IGF-2 is mediated primarily via IGF2R. Later it was demonstrated that activation of IGF2R by its another ligand, m6p, leads to similar effects [165,247]. Moreover, IGF2R is most likely responsible for clearance of protein aggregates in some neurodegenerative disorders [20,49]. The usage of specific IGF2R agonists (like Leu^27^IGF-2) when designing new medicines may help to circumvent the undesired effects on cell proliferation, since these effects of endogenous IGF-2 are mediated by IR-A [157] and IGF1R [157,163]. Specific IGF2R activation seems less likely to trigger oncogenic processes; moreover, some studies point to the role of IGF2R as a tumor suppressor [307,308]. However, there is also some data indicating IGF2R involvement in hemangioma proliferation [309].

Another approach to minimize potential side effects of peptide-based drugs on peripheral organs and tissues is intranasal administration that was shown to provide more localized increase of a drug concentration in the brain [310,311,312,313]. This administration route is also suitable for large protein molecules, like insulin, nerve growth factor (NGF) and IGF-1 [122,312]. For many substances this method allows to circumvent the BBB [313]. One of the arguments for using this route is that in *Fmr1^-/-^* mice (an animal model of Fragile X syndrome), learning and memory impairments were ameliorated by intranasal IGF-2 [101].

All the publications that we were able to discuss here show that IGF-2 is a very important molecule, with multifaceted functions in both the CNS and the peripheral tissues. IGF-2 synthesis and activity are regulated on multiple levels by complicated mechanisms (genomic imprinting; alternative promoters; mRNA-binding proteins; post-translational protein modifications; IGF-2 binding proteins in the serum; three different receptors able to bind IGF-2 with different affinities and different downstream signaling cascades). Many aspects of these mechanisms vary significantly between species. While much is already known about IGF-2 and related molecules, many details are still unclear. Future research is necessary to unveil the specifics of some cell processes controlled by IGF-2, to begin with, the exact mechanisms of IGF2R signaling transduction and the nature of factors that balance the signaling function of this receptor against its scavenging function. Two other receptors that may bind IGF-2 may be more well-studied but since most of the existing works consider other ligands of these receptors, IGF-2/IR-A and IGF-2/IGF1R interactions also deserve closer attention in the future research. Studying these mechanisms is not only interesting and useful for our understanding of normal physiology but has multiple potential clinical applications, since IGF-2 or its mimetics may prove useful as memory enhancers or neuroprotective agents, as both theoretical research and existing experimental evidence suggest. However, more suitable animal models for studying these molecules have yet to be developed, because differences between human and rodent IGF-2 systems are too numerous to use wild-type rodents in translational studies. We hope that our review would be helpful for all readers interested in this subject, including scientists who may conceive future experiments based on the present information. 

## Figures and Tables

**Figure 1 ijms-22-01849-f001:**
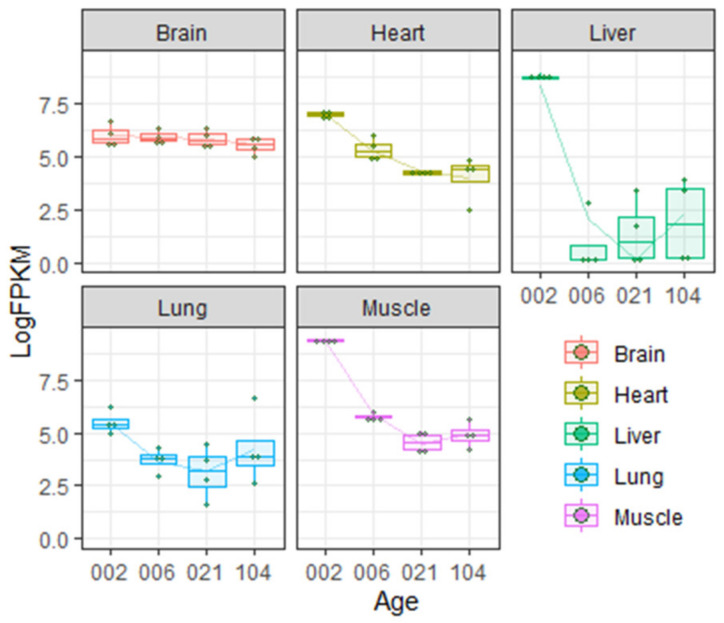
Insulin-like growth factor 2 (IGF-2) expression levels in different organs during postnatal development in rats according to Gene Expression Omnibus data repository [57,58]. Age is given in days. LogFPKM—logarithm of FPKM (fragments per kilobase of exon model per million reads mapped) value.

**Figure 2 ijms-22-01849-f002:**
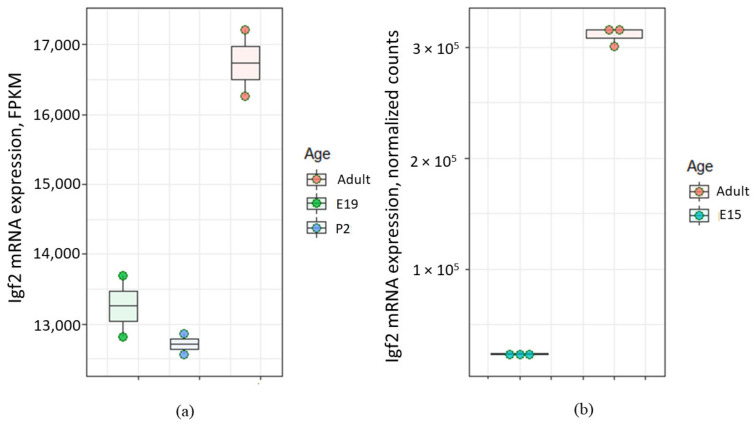
Levels of Igf2 mRNA in the choroid plexus of rats of different age according to Gene Expression Omnibus data repository. (**a**) Igf2 expression level in the choroid plexus from embryonal day 19 (E19), postnatal day 2 (P2) and adult rat according to the data from [63,64]. FPKM (fragments per kilobase of exon model per million reads mapped); (**b**) Igf2 expression level in the choroid plexus from embryonal day 15 (E15) and adult rat according to the data from [65,66].

**Figure 3 ijms-22-01849-f003:**
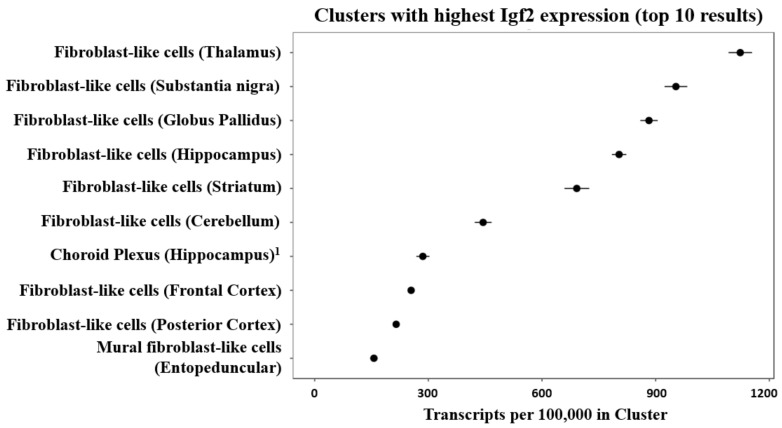
Cell clusters of the mouse brain showing the highest expression of Igf2 mRNA after single-cell sequencing from 9 mouse brain regions. The figure was generated by DropViz [68,69] and slightly modified. ^1^ It was noted that small fractions of choroid plexus were sampled from ventricle-adjacent regions including hippocampus [69].

**Figure 4 ijms-22-01849-f004:**
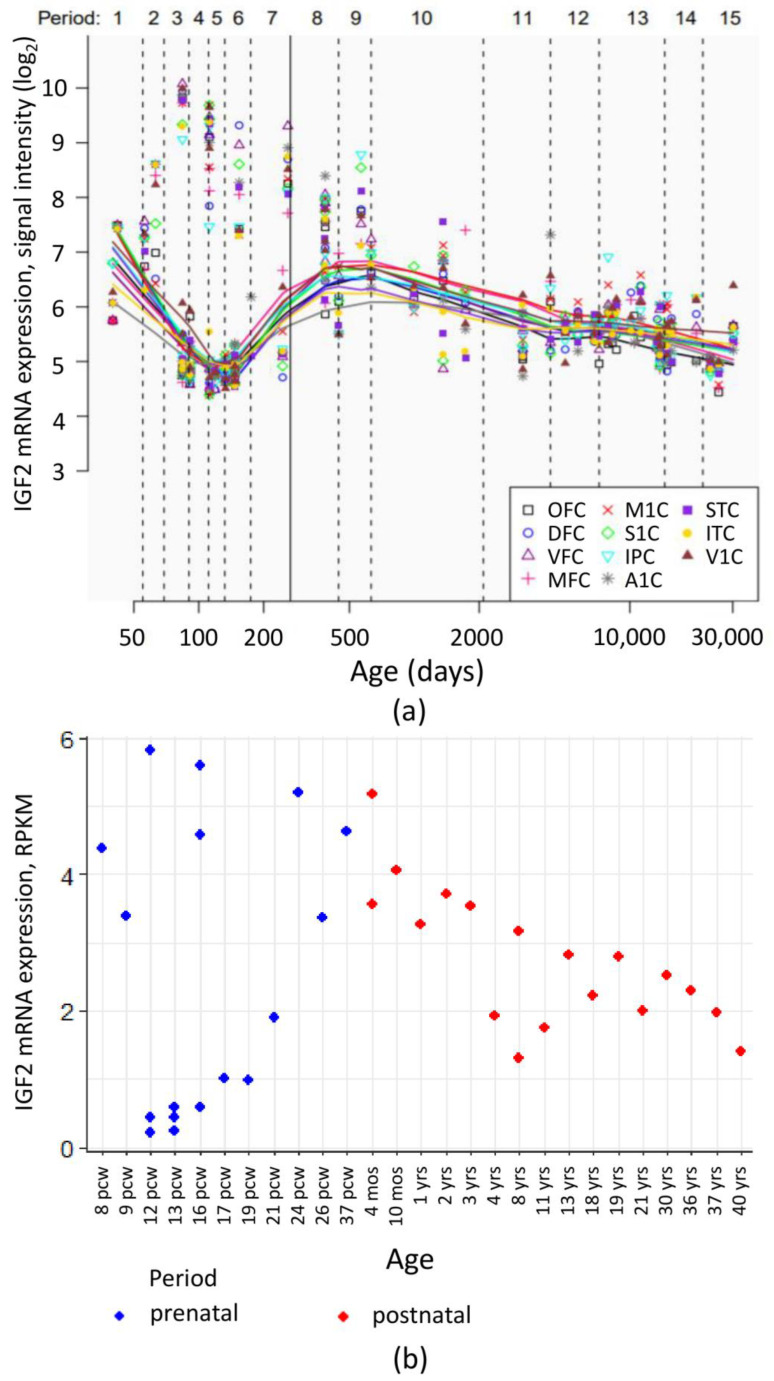
IGF2 mRNA expression levels in human cortical regions during pre- and postnatal development. (**a**) IGF2 mRNA expression levels in neocortical areas assessed in the Human Brain Transcriptome project [76,77]. OFC—orbital prefrontal cortex, DFC—dorsolateral prefrontal cortex, VFC—ventrolateral prefrontal cortex, MFC—medial prefrontal cortex, M1C—primary motor (M1) cortex, S1C—primary somatosensory (S1) cortex, IPC—posterior inferior parietal cortex, A1C—primary auditory (A1) cortex, STC—superior temporal cortex, ITC—inferior temporal cortex, V1C—primary visual (V1) cortex. Vertical line denotes the time of birth. Age is given in postconceptional days. (**b**) IGF2 mRNA expression levels (RPKM, reads per kilobase of exon model per million mapped reads) in dorsolateral prefrontal cortex during human development. Image is based on the data obtained from [78,79]. pcw—postconceptional week, mos—months of postnatal development, yrs—donor age in years.

**Figure 5 ijms-22-01849-f005:**
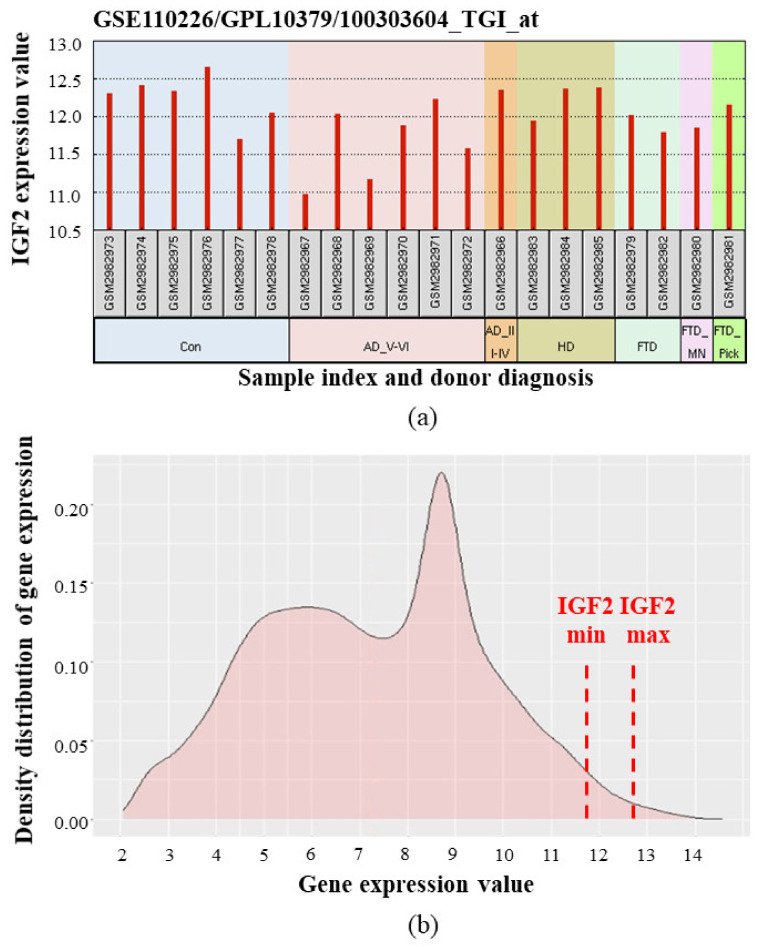
IGF2 mRNA expression in human choroid plexus according to [85,86]. Microarray data (median-centered values, normalized and cross-comparable across replicates) were processed using the GEO2R tool [87,88]. (**a**) IGF2 expression levels in choroid plexus samples from individual donors. Con—control (healthy) donors, AD_V-VI—advanced Alzheimer’s disease (Braak stage V–VI), AD_III-IV—Alzheimer’s disease (Braak III–IV), HD—Huntington’s disease, FTD—frontotemporal dementia, FTD_MN—frontotemporal dementia and motor neuron disease, FTD_Pick—Pick’s disease (a specific type of FTD); (**b**) Averaged density distribution of gene expression in choroid plexus from healthy donors. Red dash lines indicate minimal and maximal *IGF2* expression levels. *IGF2* gene is located in the far right part of the graph, meaning that *IGF2* is one of the most expressed among all the studied genes. (Analysis of density distribution of gene expression is described in detail in [89]).

**Figure 6 ijms-22-01849-f006:**
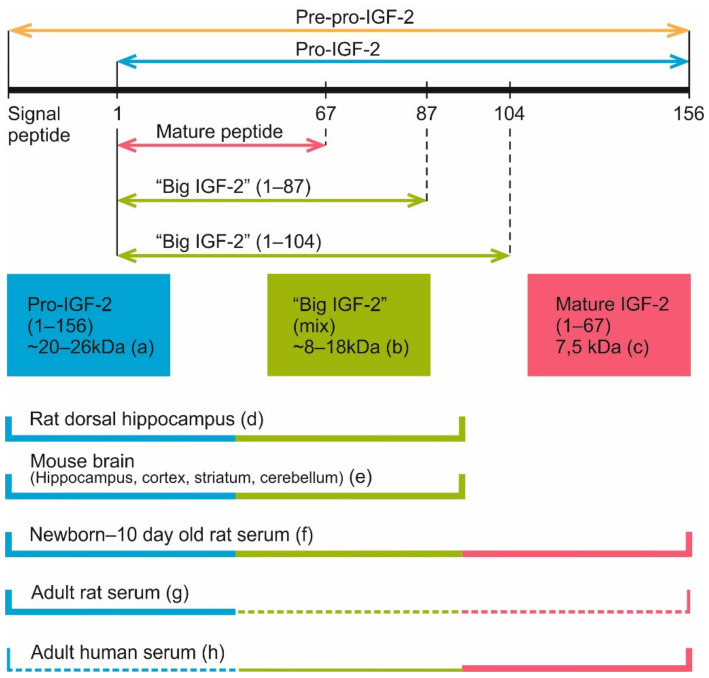
Currently known IGF-2 proteolysis sites and IGF-2 proteoforms discovered in blood serum and brain samples of different species. (**a**) According to various reports, pro-IGF-2 (1-156 aa) has a molecular weight about 20–26 kDa in different tissues [36,107,108]. (**b**) Proteoforms 1–87 aa and 1–104 aa, also known as “big IGF-2”. A few products of incomplete pro-IGF-2 cleavage weighing from 8 to 18 kDa were registered in some experiments [35,36,106,112]. (**c**) Mature IGF-2 (1-67 aa) has a molecular weight of 7.5 kDa [7,54,106,109,110]. (**d**) 15 and 20 kDa IGF-2 proteoforms were found in rat dorsal hippocampus [108]. (**e**) 14 and 21–22 kDa IGF-2 proteoforms were detected in mouse brain regions (hippocampus, cortex, striatum and cerebellum), with some additional bands being present on Western blots [111]. 17 kDa IGF-2 was found in mouse hippocampus and entorhinal cortex [20]. **(f)** Mature IGF-2, “big IGF-2” and pro-IGF-2 are present in the serum of neonatal [54] and 10 days old rats [112]. (**g**) In the serum of adult rats, pro-IGF-2 is the predominant proteoform, while mature and “big” IGF-2 are either present in low quantities [54] or not detected at all [112]. (**h**) Mature IGF-2 (7.5 kDa) is the main IGF-2 proteoform in adult human serum (plasma) [54,110,112], while “big IGF-2” is present there in low quantity [110,112]. Pro-IGF-2 was also revealed in the human serum in some studies [112] but not detected in others [54,110]. Thick lines denote proteoforms that were clearly present in samples, thin lines denote proteoforms that were reported to be present in low quantity, dash lines denote proteoforms that were detected only in some of all the referenced studies.

**Figure 7 ijms-22-01849-f007:**
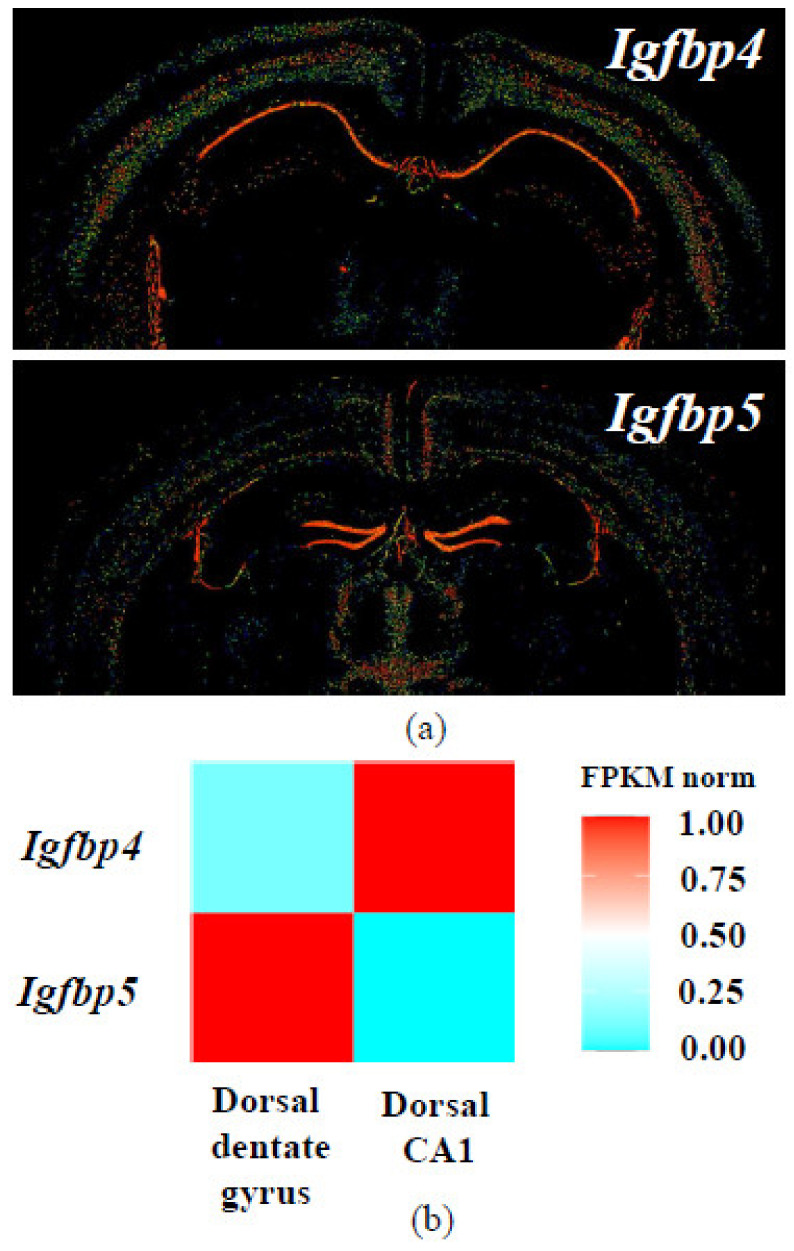
Expression patterns of Igfbp4 and Igfbp5 mRNA in the mouse brain according to databases. (**a**) In situ hybridization results from Allen Mouse Brain Atlas [144,145,146]; (**b**) Normalized Igfbp4 and Igfbp5 expression heatmap for dorsal CA1 and dentate gyrus hippocampal principal neurons from Hippocampus RNA-seq atlas [147,148].

## Data Availability

Publicly available datasets provided by third parties were analyzed in this study. These data can be found in Gene Expression Omnibus: GEO accession number GSE53960, (https://www.ncbi.nlm.nih.gov/geo/query/acc.cgi?acc=GSE53960 (accessed on 8 February 2021)), GSE44056 (https://www.ncbi.nlm.nih.gov/geo/query/acc.cgi?acc=GSE44056 (accessed on 8 February 2021)), GSE44072 (https://www.ncbi.nlm.nih.gov/geo/query/acc.cgi?acc=GSE44072 (accessed on 8 February 2021)), GSE110226 (https://www.ncbi.nlm.nih.gov/geo/query/acc.cgi?acc=GSE110226 (accessed on 8 February 2021)). We found single cell expression profiles in the DropViz database http://dropviz.org/ (accessed on 8 February 2021) and in the Hippocampus RNA-seq atlas https://hipposeq.janelia.org (accessed on 8 February 2021). Human cortex transcriptome data were obtained from the Human Brain Transcriptome https://hbatlas.org (accessed on 8 February 2021), as well as from the BrainSpan Atlas of the Developing Human Brain http://www.brainspan.org (accessed on 8 February 2021). In situ hybridization images were obtained from the Allen Mouse Brain (https://mouse.brain-map.org/experiment/show/7192431 (accessed on 8 February 2021) https://mouse.brain-map.org/experiment/show/73592530 (accessed on 8 February 2021)). All these links are listed in the References section as well.

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
