# Peer review of "Insulin-Like Growth Factor 2 As a Possible Neuroprotective Agent and Memory Enhancer—Its Comparative Expression, Processing and Signaling in Mammalian CNS"

_ijms, 2021, doi:10.3390/ijms22041849_

Round 1

Reviewer 1 Report

The authors have discussed the basics of IGF2 research among different species. They have highlighted the importance of IGF-2 as a neuroprotective agent. The review is nicely written and the topics are very well explained. More examples of the disease studies need to be explained since they mentioned this in the beginning. 

Author Response

We thank the reviewer for the thorough reading of our manuscript. According to the review #1 we added a few more examples of the changes in IGF-2 expression in various diseases in the introduction. The role of IGF2 in treating some neurodegenerative disorders in animal models is discussed in a new section “IGF-2 as a neuroprotective agent”. Also, we needed to thoroughly revise the paper as the other reviewer requested.

Reviewer 2 Report

Major Comments

Major Comments

  1. After reading this paper, it is still unclear what the potential role of IGF-2 is as neuroprotective agent. This review does not provide good arguments to show that IGF-2 is a very important neuroprotective molecule. 
  2. This review does not give a well-balanced review about IGF-2 as a possible neuroprotective agent. The paper should be shortened, more focused and informative about the topic, and more concisely written.
  3. In this review the role of IGF-2-mediated effects by the IGF-1R in neuroprotection is undervalued by the authors.
  4. The role of IGF-1 in neuroprotection is undervalued by the authors.
  5. The very long paragraph about IGF-2 receptor (6 pages) and the Discussion section lack both clear messages and conclusions.
  6. The authors present mainly data about mRNA expression of IGF-2. However, correlation between expression levels of mRNA and protein are notoriously poor.
  7. The authors discuss many studies performed in rodents and suggest that IGF-2 may be helpful in treating some neurological disorders. However, it is very questionable whether observations in rodents may be translated to humans, especially since expression of IGF-II is down-regulated after birth in rodents. In contrast to rodents, postnatal IGF2 expression in humans is driven by the liver-specific promoter P1, resulting in high postnatal serum IGF-II levels which increase until puberty. In addition, in contrast to rodents the postnatal expression of the IR-A in humans is high.
  8. For all figures: units should be mentioned at the Y- and X-axes.
  9. Figure 1. The abbreviation LogFPKM (label) should be explained.
  10. Figure 2. The abbreviation FPKM (label) should explained.
  11. Figure 3. It is unclear which parameters are mentioned at the Y-AXIS. What is fibroblast-like Dcn? Please explain [#6], [#14], [#9] etc.?

Author Response

We thank the reviewer for the thorough reading of our manuscript.

  1. After reading this paper, it is still unclear what the potential role of IGF-2 is as neuroprotective agent. This review does not provide good arguments to show that IGF-2 is a very important neuroprotective molecule.

We thank the reviewer for pointing this out. We revised the paper and added a new section that focuses on the neuroprotective role of the IGF-2, combined there some references that were previously scattered between different paragraphs and also added a few more references about this. We also decided to make a minor change to the title and add the words “memory enhancer” there because the role of IGF-2 as a memory enhancer was already discussed in the text in great detail.

  1. This review does not give a well-balanced review about IGF-2 as a possible neuroprotective agent. The paper should be shortened, more focused and informative about the topic, and more concisely written.

We shortened the section about IGF2R since this is not the main topic of our review. We also compiled the arguments about IGF-2 having neuroprotective properties together in the separate new section, and the references about IGF-2 role in learning and memory in another new section. We added more exact information about the structure of the described molecules. We tried our best to make the data in the paper more organized.

  1. In this review the role of IGF-2-mediated effects by the IGF-1R in neuroprotection is undervalued by the authors.

We added the information about the likely role of IGF-2/IGF1R signaling promoting neuroprotection (more particularly, survival of newborn hippocampal neurons in the SGZ of adult mice) according to Agis-Balboa, R.C. et al. (2011). We stated more clearly that IGF-2 effects on cell proliferation, differentiation and survival are mediated mostly via IGF1R in the section about this receptor. We also cite multiple sources about the role of both IGFs and their receptors in neuroprotection, but IGF-2/IGF1R interaction was not addressed separately in these studies.

  1. The role of IGF-1 in neuroprotection is undervalued by the authors.

IGF-1 is not the main topic of our paper, but undeniably there is a number of similarities between IGF-1 and IGF-2 function. We added a few more comparisons between these two growth factors in different paragraphs, particularly in the section about neuroprotective effects of IGF-2. We also added a few references about the connection between IGF-1, exercise, brain plasticity, neurogenesis and neuronal survival in the same section.

  1. The very long paragraph about IGF-2 receptor (6 pages) and the Discussion section lack both clear messages and conclusions.

In the revised version, we significantly shortened the paragraph about IGF2R. Mostly we removed the parts explaining the details of IGF-2-independent lysosomal function of IGF2R. However, we could not fully omit the description of this function since this is the primary function of this receptor which is very important for the cell. The information about IGF2R involvement in learning and memory was moved into a new paragraph about the role of IGF-2 as a memory enhancer. We formulated our conclusions in the paragraph about IGF2R and in the Discussion section more clearly.

  1. The authors present mainly data about mRNA expression of IGF-2. However, correlation between expression levels of mRNA and protein are notoriously poor.

In the revised manuscript, we added more information about IGF-2 protein levels in different cases and made a comparison of IGF2 mRNA transcription rate in some organs and IGF-2 concentration in the serum of rats and humans of different ages. While collecting the data for our review, we sought the information about both mRNA and protein expression of IGF-2 in all cases we would like to describe (different species, different tissues and organs, different developmental stages). However, in most sources only mRNA expression or (less often) only protein expression was measured. So unfortunately these data are incomplete. For single-cell experiments, only mRNA expression data is available since protein levels are routinely measured only in bulk tissue.

  1. The authors discuss many studies performed in rodents and suggest that IGF-2 may be helpful in treating some neurological disorders. However, it is very questionable whether observations in rodents may be translated to humans, especially since expression of IGF-II is down-regulated after birth in rodents. In contrast to rodents, postnatal IGF2 expression in humans is driven by the liver-specific promoter P1, resulting in high postnatal serum IGF-II levels which increase until puberty. In addition, in contrast to rodents the postnatal expression of the IR-A in humans is high.

In our review we repeatedly say that in the case of IGF-2 system, the differences between human and rodents on different levels of IGF-2 expression and regulation are too numerous, and that wild-type rodents should be used carefully as model organisms in translational studies. We propose two alternatives: using either another species or genetically modified rodents with altered, more human-like IGF-2 expression patterns. Both these alternatives are currently no more than speculations, and to our knowledge, no attempts to generate a specific strain of rodents expressing the mature proteoform of IGF-2 in their liver during adulthood, like humans, has been taken so far. We also changed the wording in the Discussion section to make our message about wild-type rodents being not optimal for such studies more clear. For in vitro studies, we also discussed the usage of neurons derived from human iPSCs as an alternative to rodent neuronal cultures.

In the original version of our manuscript, we have already listed a number of species-specific differences with the main focus on differences between humans and rodents: different IGF-2 expression levels in prenatal and postnatal development, different organs being the main source of IGF-2 in the organism, different numbers of alternative promoters (including human P1) and possible transcripts, different IGF-2 proteoform spectrum in the serum. In the revised manuscript we paid more attention to the papers about P1-driven transcription.

We indeed have not described the difference in IR-A postnatal expression in the original text. We thank the reviewer for attracting our attention to this fact, and we added more information about this to the revised version of the paper.

  1. For all figures: units should be mentioned at the Y- and X-axes.

We added the explanations about the units in figure legends.

  1. Figure 1. The abbreviation LogFPKM (label) should be explained.

We added the explanation as requested.

  1. Figure 2. The abbreviation FPKM (label) should explained.

In the original version, there was an explanation in the legend (fragments per kilobase per million reads), but we now changed it to the more accurate one (fragments per kilobase of exon model per million reads mapped).

  1. Figure 3. It is unclear which parameters are mentioned at the Y-AXIS. What is fibroblast-like Dcn? Please explain [#6], [#14], [#9] etc.?

We changed this picture and put the names of cell clusters identified in this source instead of the arbitrary cluster numbers used by the authors. Dcn stands for decorin, a protein used as a marker of fibroblast-like cells, but we now removed the mention of decorin to make the picture more concise.

Round 2

Reviewer 2 Report

The manuscript is somewhat improved and I have no further comments.